# Metamorphic pressure variation in a coherent Alpine nappe challenges lithostatic pressure paradigm

Cindy Luisier [1*], Lukas Baumgartner[1], Stefan M. Schmalholz[1], Guillaume Siron[1,2] & Torsten Vennemann[3]

Pressure–temperature–time paths obtained from minerals in metamorphic rocks allow the reconstruction of the geodynamic evolution of mountain ranges under the assumption that rock pressure is lithostatic. This lithostatic pressure paradigm enables converting the metamorphic pressure directly into the rock's burial depth and, hence, quantifying the rock's burial and exhumation history. In the coherent Monte Rosa tectonic unit, Western Alps, considerably different metamorphic pressures are determined in adjacent rocks. Here we show with field and microstructural observations, phase petrology and geochemistry that these pressure differences cannot be explained by tectonic mixing, retrogression of high-pressure minerals, or lack of equilibration of mineral assemblages. We propose that the determined pressure difference of 0.8 ± 0.3 GPa is due to deviation from lithostatic pressure. We show with two analytical solutions for compression- and reaction-induced stress in mechanically heterogeneous rock that such pressure differences are mechanically feasible, supporting our interpretation of significant outcrop-scale pressure gradients.

[1] Institute of Earth Sciences, University of Lausanne, 1015 Lausanne, Switzerland. [2] Department of Geoscience, University of Wisconsin-Madison, Madison, Wisconsin 53706, USA. [3] Institute of Earth Surface Dynamics, University of Lausanne, 1015 Lausanne, Switzerland. *email: cindy.luisier@gmail.com

It is commonly assumed that the tendency of rocks to flow under long-term stress allows the pressure for any depth in the Earth to be calculated assuming hydrostatic pressure distribution, as is the case in a water column[1]. Such pressure, referred to as lithostatic, is essentially a linear function of depth, because the density of rocks in the lithosphere varies only by ca. 10%. Lithostatic pressure is commonly applied as a paradigm for the reconstruction of the burial and exhumation history of rock units, such that the maximum pressure estimate for a coherent rock unit is converted into its burial depth. In the last decades, the lithostatic pressure paradigm and reports of ever-increasing pressure estimates for minor rock volumes resulted in multiple, partly fundamentally different interpretations of the geodynamic evolution of the Alps.

The Alps are an imbricate nappe stack and nappe emplacement happened in an ordered succession such that the stacked order, from top to bottom, reflects the paleogeographic position from internal to external, respectively[2,3]. The studied field area is part of the prominent Monte Rosa nappe, an internal Penninic unit of the Western Alps. It belongs to the Briançonnais domain and represents the most distal European continental crust involved in the Alpine orogeny[4] (Fig. 1a). The nappe consists of pre-Alpine paragneisses with a few intercalated metabasites and K-feldspar phenocryst-rich Permian metagranites that were metamorphosed during the Alpine orogeny[5,6]. The metagranite locally contains 10–50 m large bodies (Fig. 1c) of chloritoid, talc, and phengite-bearing metamorphic rocks. These rocks are typically referred to as "whiteschist,"[7] whereby these rocks can be deformed or undeformed despite the term "schist". This study focusses on one particular whiteschist outcrop, located in the upper Ayas valley, in Aosta (Italy), ca. 200 m north of the Refuge (mountain hut) Mezzalama. The whiteschists are not xenoliths or pieces of a subduction channel that could have been incorporated later by tectonic mixing (mélange) into the metagranite during the Alpine orogeny (see below). Instead, they were formed by metasomatic alteration of the granite by late magmatic hydrothermal fluids related to the cooling of the granite intrusion long before the onset of Alpine orogenesis[8]. The granitic protolith of the whiteschist and the late magmatic, pre-Alpine, hydrothermal nature of the fluids are confirmed by the following: field observations of a gradual transition over few metres from metagranite to whiteschist, marked by the progressive disappearance of igneous phases, demonstrating the in-situ character of the whiteschist (Supplementary Fig. 1); similarity in chemical composition shown by mass balance calculations using whole rock compositions, confirming the granitic origin of the whiteschist[8] (Supplementary Fig. 2); stable isotope compositions suggesting a late magmatic hydrothermal nature of the metasomatic fluids[5]; and the pre-Alpine age of the metasomatic alteration that is independently supported by radiogenic strontium isotope data[5]. This pre-Alpine metasomatic alteration established the chemical composition of the magnesium-rich sericite–chlorite schist and subsequent closed-system Alpine metamorphism produced the whiteschist high-pressure mineralogy. Numerous Permian, pre-Alpine, dykes crosscut the metagranite–whiteschist transition (Fig. 1c) without significant direction change or deformation, attesting that the whiteschist is not a xenolith from an Alpine subduction channel juxtaposed to the granite by Alpine-age structures. The absence of high-pressure shear zones and faults related to the whiteschist outcrop together with the hydrothermal origin of the whiteschist imply tube-like chemical alteration structures (Fig. 1b). Locally observed deformation is greenschist-facies (retrograde) and post-dates the peak-pressure stage. We did not find any evidence of an important prograde to peak-pressure deformation in these rocks.

Peak Alpine metamorphism occurred at ca. 42 Ma[4] and was followed by rapid decompression[9]. The Alpine history of the nappe has been documented by numerous studies[4,10–15]. Peak pressures were first estimated between 1.2 and 1.6 GPa[10–12], corresponding to ca. 40–60 km burial depth, assuming lithostatic pressure. The tectonic evolution of the Alpine region, including the Monte Rosa nappe, has been reconstructed based on a wealth of field observations and explained by the orogenic wedge model[3,16]. This model explains nappe formation and imbricate stacking by basal accretion and nappe exhumation by ongoing stacking, subsequent backfolding, erosion, and local extension during overall convergence. Wedge formation is mainly driven by compressive tectonic forces, with buoyancy forces playing a minor role only[16]. Subsequently, pressure estimates between 2.2 and 2.7 GPa[14,15] have been determined for the Monte Rosa nappe. However, these high-pressure rocks of >2.2 GPa are restricted to minor volumes of whiteschist within the metagranite and a few mafic boudins in the polymetamorphic basement[14,15]. The lithostatic burial depths >80 km associated with the >2.2 GPa pressure estimates for the Monte Rosa nappe were used to reject the orogenic wedge model in favour of the subduction channel model[17]. This model implies that nappes are buoyancy-driven plumes rising along a subduction channel from depths >80 km[17,18]. All the other peak pressures are lower than 2.7 GPa and are traditionally explained either by a lack of equilibration of the mineral assemblage during burial due to slow reaction kinetics or by complete retrogression of the high-pressure assemblage to lower pressure during exhumation. Obviously, the lithostatic pressure assumption is crucial to all these arguments and understanding of metamorphic pressure variations recorded in the Monte Rosa nappe—in fact in any orogeny—is essential to determine the controlling forces and mechanisms of the Alpine orogeny and to understand the rock's burial and exhumation history.

Here we show, based on field and microstructural observations, phase petrology and geochemistry, that an outcrop-scale pressure difference of $0.8 \pm 0.3$ GPa is recorded by mineral assemblages. We argue that this pressure difference reflects a transient high differential stress and local deviation from lithostatic pressure. We further compare this pressure difference with the results of two analytical solutions for compression- and reaction-induced stresses to show that the reported outcrop-scale pressure variations can indeed occur in mechanically heterogeneous rock units.

## Results

**Whiteschist phase petrology.** The peak-pressure paragenesis in the whiteschist consists of chloritoid, talc, phengite, and quartz. Retrogression commonly produced kyanite, chlorite, and white mica. Calculations using the DOMINO software package[19] in combination with the Berman internally consistent thermodynamic database[20] result in the observed assemblage, correct phase compositions, and modal abundances at 2.2 GPa for a peak temperature of 540–600 °C and a $H_2O$ activity of 1 (Fig. 2a). This result is at the lower range of the previous published peak-pressure estimates for this rock[14]. Previous studies suggested $H_2O$ activities as low as 0.6 in the whiteschist[10,14], which would influence the position of the peak mineral assemblage. Therefore, we estimated $H_2O$ activity in the whiteschist based on $H_2O$ measurement in phengite by in-situ secondary ion mass spectrometry (SIMS). Results from two whiteschist samples (Fig. 2b) show a homogeneous and high $X_{OH}$ content in phengites from whiteschist, ranging from 0.93 to 1.05. For the range of compositions measured, this high $X_{OH}$ content reflects the $H_2O$ activity (see Methods for details). This supports the assumption of $H_2O$ activity of ca. 1 and hence the 2.2 GPa estimated for the whiteschist. A significant decrease of $H_2O$ activity due to high salinities

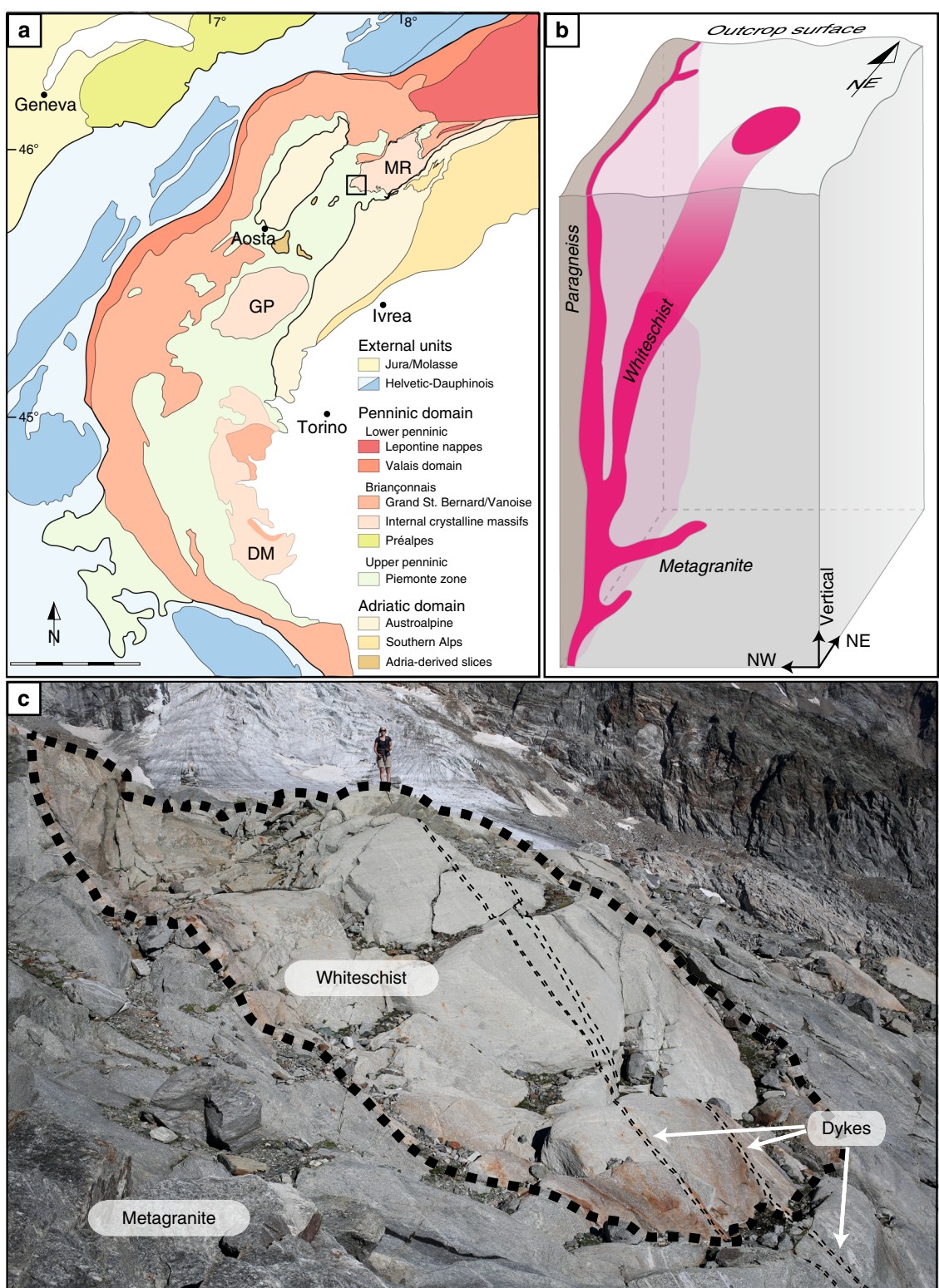

**Fig. 1** Outcrop localisation, schematic 3D geometry, and field photography. **a** Simplified tectonic map[50,51] of the Western Alps showing the location of the Monte Rosa nappe (MR). The square indicates the study area. Scale bar, 50 km. **b** Illustration of the potential 3D geometry of the whiteschist, extrapolated from 2D field data. **c** Photo of the whiteschist body within the metagranite. Pre-alpine dykes are cross-cutting both metagranite and whiteschist, indicating that the whiteschist is not an Alpine xenolith. Note the person standing on the whiteschist for scale

would result in Cl-rich phengite. Similarly, $CO_2$-rich fluids would generate magnesite or dolomite, which are not found in the primary whiteschist studied here. These observations are in agreement with the high $OH^-$ content measured in phengite.

**Metagranite peak-pressure mineralogy**. The peak metamorphic assemblage in the metagranite consists of phengite + titanite, which are partially replacing biotite, and fine-grained albite + zoisite ± phengite ± garnet intergrowth formed from igneous

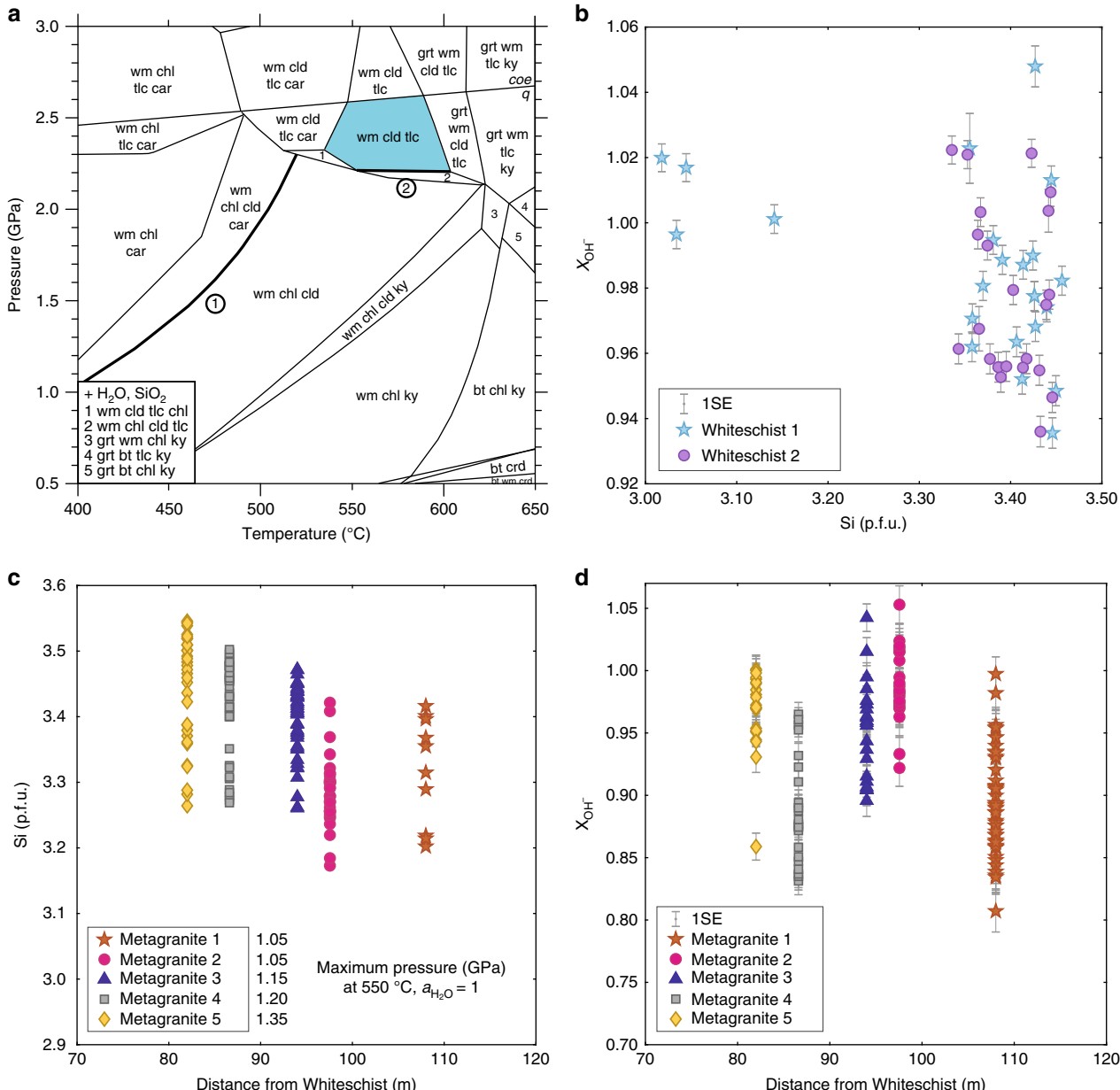

**Fig. 2** Equilibrium phase diagram of whiteschist and phengite composition in whiteschist and metagranite samples. **a** Whiteschist thermodynamic calculation in the system KFMASH, using DOMINO software and the Berman database[20] (1988, 92 update), with the peak-pressure–temperature field in blue. $H_2O$-saturated conditions were used with the following composition in moles: Si (61.79), Al (20), Fe (2.23), Mg (12.17), and K (3.82). Mineral abreviations are as follows: bt: biotite; car: carpholite; chl: chlorite; crd: cordierite; cld: chloritoid; grt: garnet; ky: kyanite; opx: orthopyroxene; tlc: talc; wm: white mica. Bold lines labelled with (1) and (2) indicate dehydration reactions considered in the discussion. **b** $OH^-$ content in phengites in two whiteschist samples vs. silica (Si) content per formula unit (p.f.u.). Error bars represent the propagated uncertainty (1 SE) of the SE of the SIMS analyses, the SE of the reference material used for SIMS analyses, and the uncertainty on the electron microprobe measurements. **c** Silica (Si) content of white mica per formula unit (p.f.u.) in metagranite is plotted vs. the distance of the corresponding metagranite from the whiteschist body. The maximum pressure for the corresponding silica content is indicated assuming a temperature of 550 °C and a $H_2O$ activity ($a_{H_2O}$) of 1. **d** $OH^-$ content in phengites in metagranites vs. the distance of the corresponding metagranite from the whiteschist body. Error bars represent the propagated uncertainty (1 SE) of the SE of the SIMS analyses, the SE of the reference material used for SIMS analyses, and the uncertainty on the electron microprobe measurements. Note that $H_2O$ activities >1 in **b** and **d** result from the accumulated uncertainty related to SIMS $H_2O$ content measurement and electron microprobe major elements measurement

plagioclase (Fig. 3a, b). Retrogression and sluggish kinetics have conventionally called upon to justify the absence of jadeite in the metagranite[11,21,22]. Jadeite should be found for pressures >1.6 GPa at a temperature of 550 °C[23]. Jadeite results from the high-pressure destabilization of plagioclase through the reaction: albite = jadeite + quartz[23]. Reaction products such as

albite, zoisite, kyanite, quartz, garnet, ±white mica start forming at a lower pressure for plagioclase compositions with small amounts of anorthite component[21]. Interestingly, jadeite has never been reported in the literature[11,12], nor have we observed it in any of the Monte Rosa metagranite in the many thin sections studied. Instead, detailed studies reveal igneous

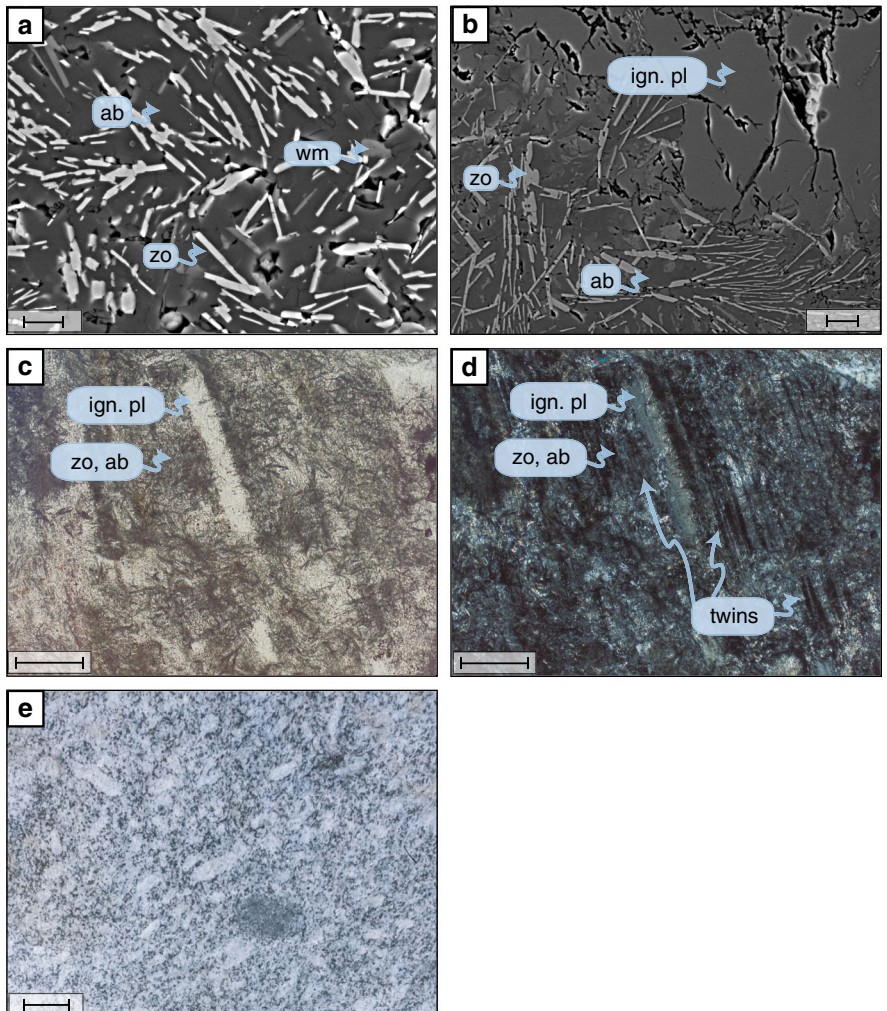

**Fig. 3** Microphotographs of high-pressure plagioclase pseudomorphs and outcrop picture of an undeformed metagranite. **a** Backscattered electron image of pseudomorphic replacement of the igneous plagioclase by fine-grained assemblage of albite, zoisite, and white mica. **b** In some places, the igneous plagioclase is partially preserved. Mineral abbreviations are as follows: ab = albite; ign. pl = igneous plagioclase; wm = white mica; zo = zoisite. **c** Plane-polarised light microphotograph of a thin section showing plagioclase pseudomorphs (ab and zo: dark fine-grained crystals) partially replacing igneous plagioclase. **d** Same area as **c**, under crossed polarisers. Note that the matrix albite in the fine-grained replacement texture has inherited the crystallographic orientation as evidenced by the twin planes. This confirms direct replacement of igneous plagioclase by the albite, confirming that jadeite never formed. **e** Picture of a typical undeformed metagranite with large K-feldspars in a matrix of plagioclase, quartz, and biotite. Scale bars are **a** 10 μm; **b** 20 μm; **c** 200 μm; **d** 200 μm; **e** 5 cm

lamellar twinning, which is typical for plagioclase in granites, in both the unreacted portions of the igneous plagioclase, as well as in the albite pseudomorphs (Fig. 3c, d). The mimicking of the igneous twinning by the albite is evidence that albite is a direct replacement phase of the igneous plagioclase, as albite inherited the same crystallographic orientation to preserve the twins. If albite was a reaction after jadeite, the crystallographic orientation of the igneous plagioclase would have been lost during jadeite growth and could hence not be inherited by albite (see similar example from ultramafic rocks[24]). The preservation of grains of igneous plagioclase and igneous twins indicate that albite is a prograde high-pressure product after plagioclase destabilization. The presence of $H_2O$-bearing minerals such as zoisite and white mica in these plagioclase pseudomorphs indicates that at least some $H_2O$ was available during the prograde reaction[25]. Hence, sluggish kinetics seems unlikely as the reason for the absence of jadeite. Overall, our textural observations suggest that the granite was never exposed to pressures significantly higher than ca. 1.6 GPa.

**Silica in phengite barometry for metagranite**. Independent pressure estimates can be made using the silica (Si)-in-phengite barometer. Pressure was determined using the calibration of Massonne and Szpurka[26] in five metagranite samples along a profile towards one of the whiteschist bodies (Fig. 2c), based on the same method (DOMINO software, with Berman thermodynamic database), which has been used for the whiteschist thermodynamic calculations. Phengites from buffered textural domains were selected, which contain the assemblage K-feldspar, quartz, biotite, and phengite. The spatial scale of chemical equilibration of phengite between biotite and the quartz, K-feldspar, and plagioclase matrix is assumed to be large enough for phengite to reach equilibrium during the high-pressure stage with the buffering mineral assemblage. The presence of titanite together with phengite and garnet in the matrix at a large distance from the biotite suggest that diffusion was efficient at least between biotite and the matrix. The biotites are not or rarely armored with coronas, as would be expected in a rock with a large amount of disequilibrium. The Si-content in phengites was determined for

various pressures experimentally by Massone and Schreyer[27]. Metagranites closer than 80 m to the whiteschist have not been analysed, as their high-pressure assemblage was partially altered during a late, after peak-pressure, greenschist-facies retrogression locally associated with deformation (see discussion above). The resulting maximum Si-content of phengites in metagranites increases from 3.42 to 3.54 Si per formula unit (p.f.u.) in the metagranite towards the whiteschist (Fig. 2c). Between 4 and 11 buffered areas per metagranite sample were considered for phengite composition measurement, corresponding to a total number of phengite analyses as follows: metagranite 1, 329 analyses; metagranite 2, 155; metagranite 3, 61; metagranite 4, 66; and metagranite 5, 52. The phengite domains around biotite consist of several phengite sub-grains, intergrown with fine-grained titanite. Several profiles were measured from the most internal position within the phengite rim towards the exterior. In each sample, the maximum Si content in phengite obtained for each analyzed area was reproducible between most areas. This reproducibility suggests that the highest Si content is most probably not missed in the analyses and, hence, that the increase in Si content in metagranites towards the whiteschist is most probably not a statistical artefact. Assuming a $H_2O$ activity of 1 and a peak temperature of 550 °C, in accordance with whiteschist estimates as well as the average peak temperature from the

literature[4,10–15], pressure increased consistently towards the whiteschist from 1.05 to 1.40 GPa. These pressure estimates are consistent with the absence of jadeite in the metagranite. Bell-shape profiles of Si-content across phengite domains (Fig. 4) show an increase in Si-content, then a "flat maximum" and subsequent decrease, from the outside towards the biotite, which was replaced by phengite during prograde reaction. We interpret the Si-content profile to reflect part of the prograde (increase in Si-content towards biotite), peak (Si-content maximum) and retrograde (decrease of Si-content towards biotite) pressure conditions of the P–T path, similar to the proposition of Evans and Patrick[28], which is based on other examples, such as Chopin and Maluski[29], and Lardeaux et al.[30]. A similar spread in Si range is interpreted to be a record of the geological evolution of the terranes from high P–T to low P–T conditions[28]. Therefore, we argue that specific silica contents reflect equilibrium at specific times in the P–T evolution during the prograde, peak, and retrograde history. We suggest that the maximum phengite Si p.f.u. corresponds to the highest-pressure conditions reached during the reaction. The bell-shape of the profile of Si-content, a flat maximum with decreasing Si-content away from the top, does not support a retrogression event related to a pervasive fluid-flush event during exhumation at a pressure of ca. 1.4 GPa, as for such a retrograde fluid-flush event a profile with homogeneous

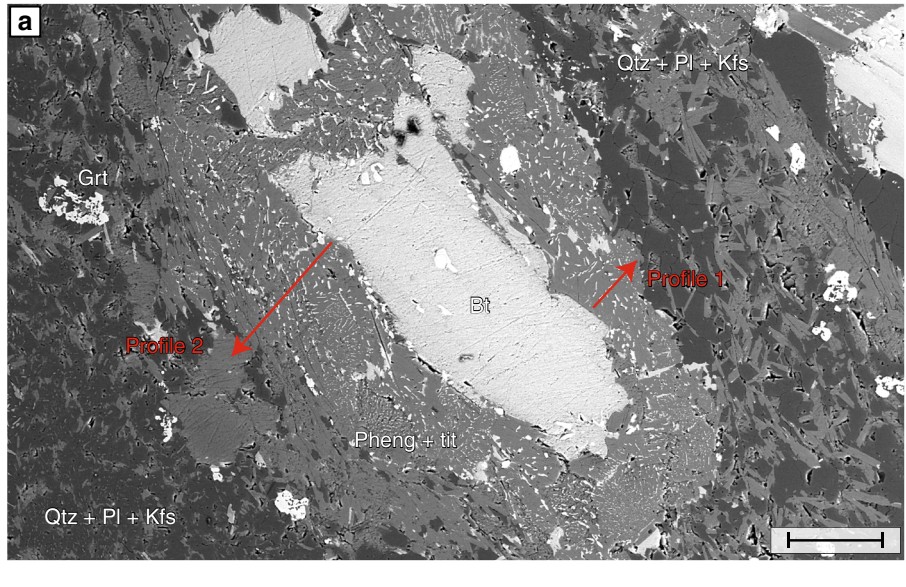

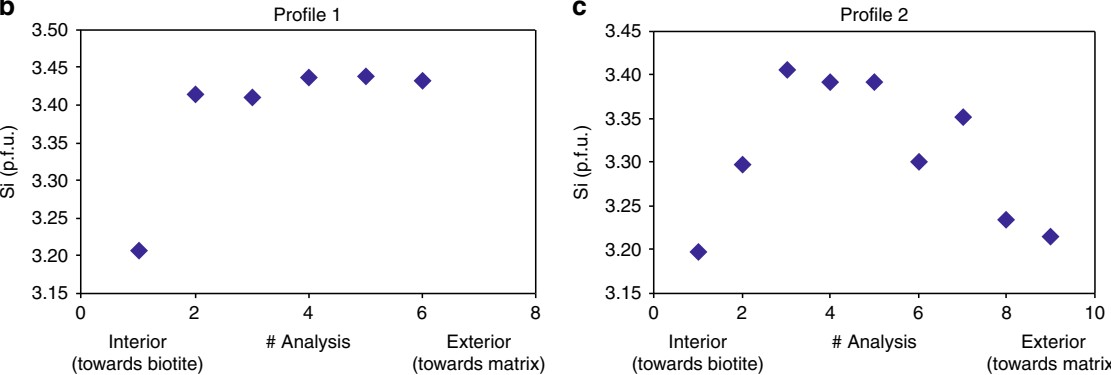

**Fig. 4** Microphotograph and compositional profiles of phengite rim in undeformed metagranite. **a** Backscattered electron image showing phengite rim around biotite. Scale bar, 100 μm. **b**, **c** Spatial variation of silica (Si) per formula unit (p.f.u.) in phengite for two profiles from a metagranite sample indicated in **a**. Profiles were measured in the reaction rims around biotite, from interior towards the matrix. The values of silica show a bell shape. The variations are interpreted to document prograde replacement of biotite by phengite, from the rim towards the centre, with subsequent pressure decrease during the retrograde phase, close to the biotite. Error bars on the silica are smaller than the symbol size

Si-content should be recorded. Moreover, a retrograde fluid-flushing event at a pressure of ca. 1.4 GPa would lead to a full breakdown of talc in the whiteschist (Fig. 2a), which is not observed. Therefore, we argue that there was no fluid-flush event after the peak pressure, which would significantly modify the peak-pressure Si-content, and that the measured Si-content represents the peak-pressure conditions. The decrease in Si content in the metagranite samples away from the whiteschist cannot be explained by metasomatic enrichment of the metagranite in silica, because according to mass balance calculations performed on metagranite and whiteschist by Pawlig and Baumgartner[8], Si is not added to the rock during metasomatism (Supplementary Fig. 2).

**H₂O content measurement in phengite.** It is well established that phengite content in white mica depends not only on pressure and temperature but also on H₂O activity[27]. For a fixed phengite composition, the calculated pressure increases with decreasing H₂O activity. The influence of H₂O activity variation in white mica on pressure for two fixed Si contents was calculated, corresponding to the maximum values in phengites in the farthest and the closest metagranite samples (3.42 and 3.54, respectively; Fig. 5). H₂O activities as low as 0.5 are still not sufficient to reach a pressure of 1.6 GPa required for jadeite appearance and pressures are still significantly smaller than the minimum pressure of 2.2 GPa determined for whiteschist. In order to estimate the actual H₂O activity during equilibration of the phengites, their H₂O content was measured by SIMS (Fig. 2d). The results show that all samples have $X_{OH}$ contents in the range of 0.81 to 1.05 in the phengites, with no systematic trend with respect to location in the crystals, or with distance to the whiteschist. This $X_{OH}$ content directly reflects the H₂O activity (see Methods for an in-depth discussion on OH⁻ white mica activity and H₂O activity relationship). It is noteworthy that the calculated OH⁻ content of the

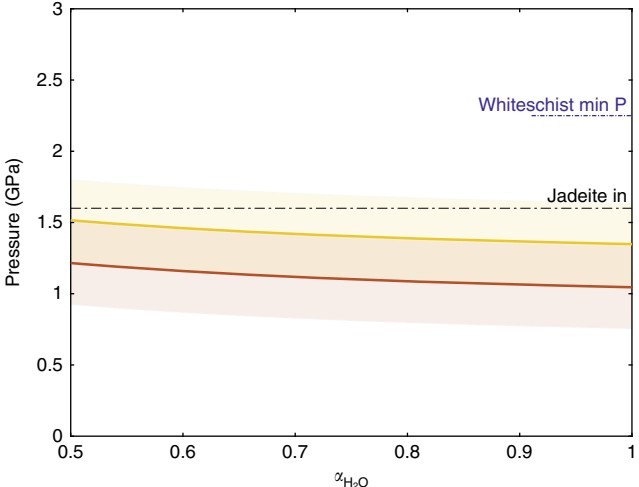

**Fig. 5** Effect of a reduction of H₂O activity on pressure for two fixed phengite silica contents of 3.42 Si p.f.u (red line) and 3.54 Si p.f.u (yellow line) assuming a constant temperature of 550 °C. Calculations and error envelopes (red and yellow fields, respectively) are based on experimental data[26,27]. The blue dashed line corresponds to the minimum peak-pressure estimate for the whiteschist (see Fig. 2a) and the black dashed line corresponds to the pressure of the jadeite in reaction[23] (calculated using Berman, 1988 database[20]). The yellow and red curves represent the pressure estimate for the metagranite closest (yellow) and farthest (red) to the whiteschist. Extremely low H₂O activities of 0.5 would still be insufficient to provide pressure estimates >1.6 GPa within the jadeite stability field

phengites from metagranite samples are considerably more variable than in the whiteschists, opening up the possibility that an actual fluid phase might have been absent in metagranites, and suggests that the H₂O content in phengite is indeed dependant on the H₂O activity on the grain boundaries. Such thin-section scale disequilibrium has previously been observed for OH⁻ content in biotite in a contact aureole environment with similar temperatures[31]. Hence, the increase of Si in phengite from the farthest to the closest metagranite from the whiteschist represents an increase in pressure and not an artefact due to H₂O activity variations. Pressure values along the sampled profile are increasing towards the whiteschist body and a pressure variation from 1.05 to 1.35 GPa over a distance of 30 m is determined in the metagranite (Fig. 2c). H₂O activities close to unity argue against an influence of mineral reaction kinetics on metamorphic pressure determinations.

## Discussion

The presented results indicate that the metagranite never experienced the peak pressure of ca. 2.2 GPa estimated for the whiteschist. According to our results, there was a metamorphic pressure difference at peak Alpine conditions between the metagranite and the whiteschist of ca. 0.8 GPa. Uncertainty of the estimated pressure difference can be evaluated by propagating the uncertainty of the pressure determination in the metagranite and the uncertainty of the pressure estimate in the whiteschist. The uncertainty of the pressure estimate in the metagranite is related to the experimental determination of the reaction albite = jadeite + quartz, corresponding to ± 0.05 GPa[23] and the uncertainty on the Si in phengite barometer, which is ± 0.03 GPa[27]. The difference in the modelled positions of reactions in whiteschists when using different thermodynamic databases results in an uncertainty of ± 0.2 GPa[20,32,33]. The propagation of uncertainties provides a final uncertainty of each pressure estimate for metagranite and whiteschist of ± 0.2 GPa, which yields an uncertainty of the pressure difference of ± 0.3 GPa ($\sqrt{0.2^2 + 0.2^2}$ GPa). A pressure difference of 0.8 ± 0.3 GPa indicates a significant deviation from the lithostatic pressure, because metagranite and adjacent whiteschist were always at the same depth during the Alpine orogeny.

Deviation from lithostatic pressure can result from differential stress in the rock and has been suggested because of the expected locally high differential stress (>100 s MPa) in a compressed lithosphere[34–36]. Recently, a correlation of a wide range of (ultra) high peak pressures with their associated pressures after decompression[36] suggest that rock pressure can indeed significantly deviate from the lithostatic pressure. Furthermore, a detailed petrological and geochemical analysis of a crustal shear zone[37], the nearly identical age and temperature of amphibolite and eclogite facies shear zones[38], the observation of grain-scale pressure variation resulting from heterogeneous stress fields[39], as well as microscale observations of the conservation of zonation in minerals[40,41] all suggest significant deviation from the lithostatic pressure in the order of several kilobars.

To test whether deviations from the lithostatic pressure in the order of 0.8 ± 0.3 GPa are mechanically feasible, we apply a simple two-dimensional analytical model in which the whiteschist is mimicked by an ellipse surrounded by a mechanically more competent material representing the metagranite (Fig. 6). The whiteschist formed from a metasomatically altered granite, resulting in the shape of a tube in three dimensions, which reflects the original metasomatic fluid pathway. Therefore, the elongated ellipse represents a section including the long axis of the tube (plane NW-vertical in Fig. 1b) and not a section more or less orthogonal to the tube (plane NW-NE in Fig. 1b), which is

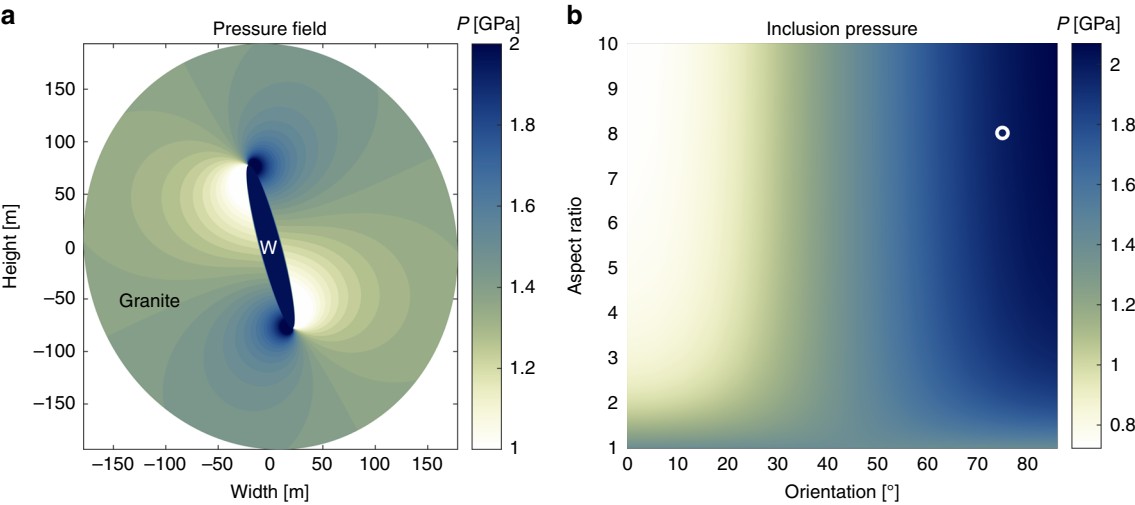

**Fig. 6** Analytical solution of pressure distribution between a weak inclusion and a strong host. Calculated pressure ($P$; using solution of ref. [45]) for a mechanically weak elliptical inclusion within a medium that is 100 times mechanically more competent. The model configuration represents a NW-vertical section of Fig. 1b, with width referring to NW and height to the vertical direction. **a** Colourplot of pressure field for a representative configuration. The pressure in the elliptical inclusion is homogeneous. The aspect ratio of the ellipse is 8. Compression direction is horizontal (parallel to width direction) and at an angle of 75° with respect to the ellipse long axis. Far-field compressive stress has both a mean and differential stress of 1.4 GPa. The elliptical inclusion (labelled W) mimics the whiteschist and the surrounding medium (labelled Granite) the stronger metagranite. **b** Colourplot of pressure, $P$, in the inclusion as a function of inclusion aspect ratio and orientation of inclusion long axis with respect to compression direction. The white circle indicates the configuration displayed in **a**. Orientations between ca. 70 and 90°, and aspect ratios >ca. 4 generate inclusion pressures of >ca. 2 GPa

observed in the field. The pressure in and around the weak elliptical whiteschist inclusion can be estimated with an analytical solution based on continuum mechanics, which is valid for plane strain and for both elastic and viscous deformation (see Methods for details)[42,43] (Fig. 6). Pressure is considerably higher in the whiteschist inclusion if it has an elongated geometry (aspect ratio >ca. 4; Fig. 6b) and is compressed at a high angle (>ca. 70°) with respect to its long axis[43] (Fig. 6b). If, e.g., a far-field compressive total stress is applied with a mean stress of 1.4 GPa (centre of Mohr stress circle), corresponding to the thermodynamic pressure in the metagranite, and a differential stress of 1.4 GPa (diameter of Mohr stress circle), then the maximal principal stress, $\sigma_1$, is 2.1 GPa. For this stress state we consider, as illustrative example, a compression direction of 75° with respect to the inclusion's long axis so that the pressure (mean stress) in the inclusion is ca. 2 GPa, close to $\sigma_1$, and the pressure in the surrounding medium is ca. 1.4 GPa (Fig. 6). A similar inclusion model has recently been applied by Jamtveit et al.[38] to explain the pressure differences between shear zones. Given the irregular shape and orientation of the tube-like whiteschist body, it is impossible to know the exact orientation of the tube with respect to the direction of the compressive force at peak-pressure conditions. However, the whiteschist–metagranite transition is subvertical in the outcrop surface (Fig. 1c), and the structures associated with the nappe formation and exhumation have a subhorizontal dipping. Therefore, it is possible that the tube axis was locally at high angle (>60°) to the compressive stress, as considered in our inclusion model. Significant mechanical strength of the metagranite during peak-pressure conditions can be justified, because field observations show that large volumes of the metagranite have remained essentially undeformed during peak-pressure Alpine deformation (Fig. 3e) and observable deformation in the metagranite is local and mostly retrograde after peak pressure. In contrast, the surrounding Variscan basement was significantly deformed during Alpine deformation[6]. Also, the abundance of centimetre-scale feldspar crystals (Fig. 3e) suggests that the effective compressive strength of the metagranite is not

dominated by quartz but more likely by the stronger feldspar (Supplementary Fig. 3)[44]. The spatial pressure variation, calculated with the analytical model, within the host metagranite at distances between 80 and 110 m away from the inclusion can correspond to a spatial variation of ~0.2 GPa over 30 m (Fig. 6a). The pressure variation is either positive or negative away from the inclusion, depending on the profile orientation. The spatial pressure variation determined in the natural metagranite profile between 80 and 110 m away from the whiteschist tube is 0.3 GPa over 30 m (Fig. 2c). Therefore, the spatial pressure gradient determined in the host metagranite could well reflect heterogeneous pressure distribution in the strong host. In the model, the pressure (mean stress) in the weak inclusion is significantly larger than the average pressure in the metagranite but the differential stress in the weak inclusion is small (<0.1 GPa). Phase equilibria models are based on data from experiments performed under hydrostatic stress, i.e., no differential stress. The mean stress in rocks with small differential stress is a good proxy for the thermodynamic pressure used in phase equilibria models[45]. Consequently, the modelled mean stress in the weak inclusion can be compared with the pressure estimate for the whiteschist based on phase equilibria models.

The above model can explain pressure differences between strong metagranite and weak whiteschist with an externally applied compressive differential stress in the metagranite. Another possibility to generate pressure differences between whiteschist and metagranite is due to dehydration reactions occurring during prograde metamorphism in the whiteschist and the related volume changes. The resulting volume changes and associated reaction-induced stresses could lead to a deviation from the lithostatic pressure in the whiteschist. For example, we assume an undeformable and impermeable metagranite host, so that fluids released during dehydration of the whiteschist cannot escape. Also, as the surrounding metagranite is essentially undeformable, the total volume change, $\Delta V_{tot}$, of the whiteschist is zero during dehydration (isochoric reaction). $\Delta V_{tot}$ is the overall volume of reaction, composed of the sum of the volume

change due to the reaction releasing fluids, $\Delta V_r$, and the volume change due to elastic compressibility, $\Delta V_e$, of the whiteschist. All volume changes are normalised by a reference volume, $V_0$. The pressure change, $\Delta P$, due to elastic compressibility is quantified by a compressibility, $\beta_0$, according to $\beta_0 \Delta P = -\Delta V_e/V_0$. For $\Delta V_{tot} = 0$, the $\Delta P$ can then be related to $\Delta V_r$ by $\Delta P = \Delta V_r/(\beta_0 V_0)$ (see Methods for details). Two major prograde dehydration reactions take place in the whiteschist as follows[46]: (1) at constant pressure and increasing temperature, carpholite breakdown and appearance of chloritoid, through the breakdown reaction of carpholite to chloritoid; (2) at constant temperature and increasing pressure, chlorite breakdown and appearance of talc (Fig. 2a). The pressure change, $\Delta P$, is calculated for reaction (1) between 400 °C, 1.5 GPa and 500 °C, 1.5 GPa, and for reaction (2) between 560 °C, 2.0 GPa and 560 °C, 2.3 GPa. A pressure increase, $\Delta P$, of 0.4 GPa (reaction 1) and 0.5 GPa (reaction 2) is generated by the dehydration reaction and this reaction-induced pressure increase in the whiteschist could also contribute to the pressure difference between whiteschist and metagranite. This model assumes that shear stresses in the metagranite, caused by the pressure increase in the whiteschist, are below the yield stress so that fracturing in the metagranite does not occur, which is in agreement with field observations. The above two simple mechanical models show that pressure differences between metagranite and whiteschist in the order of 0.8 ± 0.3 GPa could result from externally applied compressive differential stress and/ or reaction-induced stress related to dehydration reactions.

The result of our study suggests that the peak pressure ≥2.2 GPa in the whiteschist cannot be directly transformed into a burial depth for the entire Monte Rosa nappe assuming lithostatic pressure. Indeed, the peak pressure of ca. 1.4 GPa determined in the metagranite would indicate a maximal burial depth of 40–60 km of the Monte Rosa nappe, which is a burial depth in agreement with field-based structural reconstructions and the orogenic wedge model[3,16]. However, it is the particular three-dimensional stress field that determines which of the three principal stresses is close to the lithostatic pressure[47] and, hence, is a burial depth indicator. In any case, our results indicate that the maximal burial depth of the Monte Rosa nappe was less than the >80 km depth, which would correspond to a lithostatic pressure of >2.2 GPa in the whiteschist. We speculate that a high differential stress in the metagranite was transient in time and occurred during the compressive stage of nappe initiation, which is at the onset of detaching the metagranite and surrounding Variscan basement from the subducting European crust. The duration of the high differential stress episode was supposedly short-lived (≪1 Ma) and, considerably shorter than the burial–exhumation cycle of the Monte Rosa nappe. Hence, the metagranite remained mostly undeformed and differential stress was likely viscously relaxed during exhumation[48]. We also speculate that dehydration reactions and associated reaction-induced stress in the whiteschist could have contributed to the pressure difference between metagranite and whiteschist. The presented microstructural, petrological, and geochemical analyses do not support an Alpine state of lithostatic pressure in the Monte Rosa nappe. A shift of the lithostatic pressure paradigm, which is currently one of the fundamental pillars for geodynamic reconstructions, towards dynamic pressure in collisional orogens seems more appropriate to explain the observed pressure variations in the Monte Rosa nappe.

## Methods

**Analytical methods**. Major and minor element compositions of white mica were obtained using a JEOL 8200 Superprobe EPMA and a JEOL JXA-8530F HyperProbe EPMA at the University of Lausanne, Switzerland. Operating conditions were 15 kV and 15 nA. Natural minerals were used as reference material. Structural formulae were calculated on the basis of 11 oxygens. SIMS analyses were made with a CAMECA IMS 1280 at the SwissSIMS facility of the University of Lausanne, Switzerland. White mica reference materials consisting of natural white micas from various compositions between the muscovite–celadonite end members. They were made for $H_2O$ measurements[49]. A 0.5 nA $Cs^+$ primary beam at 10 kV was used and a spot diameter of 10 μm was achieved. The absolute $H_2O$ weight percent of the reference materials were obtained using high Temperature Conversion Elemental Analyzer (TC/EA) at the Stable Isotope Laboratory of the University of Lausanne. Cl and F contents of the reference materials and Monte Rosa phengites were measured using a JEOL JXA-8530F HyperProbe EPMA at the University of Lausanne.

The structural $OH^-$ content was calculated on the basis of 11 oxygens in the structural formula. The mole fraction of $H_2O$ in the white mica is calculated using the $OH^-$ concentration p.f.u. divided by 2. Backscattered electron images were acquired on a Tescan Mira II LMU field emission SEM at the University of Lausanne.

**Thermodynamic calculations**. Equilibrium phase diagram was calculated using DOMINO software[19] in combination with the internally consistent and entirely experimentally derived Berman thermodynamic database[20] (1988, 92 update) in the simple KFMASH system. The initial whiteschist composition is (in wt%): $K_2O$: 3.02, $Na_2O$: 0.15, CaO: 0.22, FeO: 1.62, $Fe_2O_3$: 2.39, MgO:8.24, $Al_2O_3$: 2.39, $TiO_2$: 0.47, $SiO_2$: 62.38, LOI: 4.02. The oxides of $Na_2O$, CaO, and $TiO_2$ were not considered, as they occur in small amounts (<0.5 wt%). All iron was considered as ferrous. Talc activity was reduced and fixed at 0.86 in the database, according to measured talc composition at the estimated peak and assuming ideal molecular mixing. The solution models used are Massonne and Szpurka[26] for white mica, whereas the chloritoid, garnet, carpholite, chlorite, and biotite solution models are from Berman[20] 1988, 92 update. Silica in phengite from metagranites is modelled using the Berman database (1988, 92 update) and the DOMINO software with the white mica solution model of Massonne and Szpurka[26] for consistency with the whiteschist calculations. The thermodynamic calculations for the metagranite result in essentially two fields, the plagioclase granite at low pressure and the jadeite granite at high pressure, separated by the plagioclase breakdown reaction. Silica isopleths are modelled at peak temperature of 550 °C (average temperature from published estimates[4,10–15]). Input composition is a bulk metagranite with (in wt%): $K_2O$: 4.57, $Na_2O$: 2.95, CaO: 1.69, FeO: 1.61, $Fe_2O_3$: 1.25, MgO: 0.85, $Al_2O_3$: 14.24, $TiO_2$: 0.48, $SiO_2$: 70.83, MnO: 0.05, $P_2O_5$: 0.20, $H_2O$: 1.08. CaO, $P_2O_5$, MnO, and $TiO_2$ were set to 0 and iron is considered as all ferrous. $H_2O$-saturated conditions were used.

**Estimation of $H_2O$ activity**. We present data on the hydroxyl composition of phengite in the whiteschist and the metagranite. The larger variability of the $OH^-$ content of phengite suggests that $H_2O$ activity on the grain boundary could be estimated through the reaction $H_2O$-phengite = > oxy-phengite + $H_2O$. Nevertheless, the thermodynamics of this reaction is unknown, such that only qualitative and relative estimates can be made. Assuming the above reaction, we can write for two different phengites in equilibrium with two different fluids: $\mu_{H_2O}^{phen1} = \mu_{H_2O}^{fl1}$ and $\mu_{H_2O}^{phen2} = \mu_{H_2O}^{fl2}$, where phen1 and phen2 denote the two different phengites, fl1 and fl2 the different fluids, and μ stands for the chemical potential. Expanding the chemical potential of $H_2O$ for each phase by introducing the activity of the component, $\mu_{H_2O} = \mu_{H_2O}^{\circ,i} + RTln\,a_{H_2O}^i$, where R is the universal gas constant, $T$ the temperature, $a_{H_2O}^i$ the activity of $H_2O$ in the phase $i$ (phen1, phen2, fl1 or fl2), and $\mu_{H_2O}^{\circ,i}$ is the chemical potential of the pure $H_2O$ fluid or pure endmember $H_2O$ phengite. Hence, the activity of $H_2O$ in fluid 2 can be related to that in fluid 1 (at constant pressure and temperature) by:

$$a_{H_2O}^{fl2} = a_{H_2O}^{fl1}\, a_{H_2O}^{phen2}/a_{H_2O}^{phen1} \qquad (1)$$

Assuming Raoult's law is valid for both phengites, the activities of $H_2O$ in the phengites can be replaced by the mole fractions of $H_2O$ in the structural formula of these phengites. A reference value of $a_{H_2O}^{fl1}$ has to be obtained so that Eq. (1) is useful. In the absence of thermodynamic data, we used the assumption that $a_{H_2O}^{phen1}$ is close to 1 for a pure $H_2O$ fluid phase, where $a_{H_2O}^{fl1} = 1$. A further challenge arises from the fact that we propose pressure gradients in the metagranites. In fact, Eq. (1) is only useful if the standard state chemical potentials refer to the same pressure and temperature. In addition, the above derivation requires that the chlorine (Cl) and fluorine (F) concentrations are small in comparison to the observed hydroxy-concentration changes. In the present case they are 0–0.31 mol% and 0–1.19 mol% for $Cl^-$ and $F^-$, respectively, in comparison with $OH^-$ variations of 12.25 to 16.24 mol%. Quantitative analysis of $OH^-$, $Cl^-$, and $F^-$ sums up to <2 in the white mica structural formula. This suggests the presence of $O^{2-}$, substantiating the above reaction involving oxy-phengite.

We measured phengite compositions, which have completely filled $OH^-$ sites in the whiteschists, where $H_2O$ activity is assumed to be close to unity.

**Analytical solution for pressure field**. The applied analytical solution[42] is based on Muskhelishivili's method that uses complex variables to solve the governing bi-harmonic equations. The solution is related to standard engineering solutions for stress concentrations around holes in incompressible material. Such analytical solutions were originally derived for elastic materials but the solutions for viscous material are mathematically identical if the shear modulus is replaced by a viscosity and the strain by a strain rate. The applied analytical solution quantifies the stress and pressure variation in and around an elliptical inclusion in a surrounding medium in two dimensions. The inclusion can be mechanically stronger or weaker than the surrounding medium. For elastic material the strength ratio between inclusion and surrounding medium is determined by the ratio of the corresponding shear moduli whereas for viscous material the strength ratio is controlled by the corresponding ratio of the viscosities. The solution is scale independent and hence applies to inclusions of any size in the framework of continuum mechanics. A far-field horizontal compressive stress state is applied to the medium and this stress state is composed of a mean stress (negative pressure) and a differential stress.

**Fluid-overpressure calculations**. Thermodynamic data were obtained with Theriak software[19] and, for consistency, again with the Berman database (1988, 92 update)[20], using the input composition of whiteschist used for equilibrium phase diagram calculations in Fig. 2a. Volume ($V$) and density ($\rho$) of solids and fluid were calculated for specific pressure ($P$) and temperature ($T$) conditions. $\Delta V_r$ is calculated as $V_1 - V_0$ and compressibility ($\beta_0$) is calculated as $\beta_0 = (1/\rho_{0^*}) (\rho_{0^*} - \rho_0)/(P_{0^*} - P_0)$ with $\rho_0$ and $\rho_{0^*}$ being densities at two different pressures $P_0$ and $P_{0^*}$, respectively, in the same stability field. Subscripts $0$ and $0^*$ stand for the reactant side of the reaction and subscript $1$ stands for the product side of the reaction.

$V_0$, $V_{0^*}$, $V_1$ and $\rho_0$, $\rho_{0^*}$, $\rho_1$ for reaction 1 were calculated at 400 °C, 1.5 GPa ($P_0$), 400 °C, 1.6 GPa ($P_{0^*}$) and 500 °C, 1.5 GPa ($P_1$) and for reaction 2 at 560 °C, 2.0 GPa ($P_0$), 560 °C, 2.1 GPa ($P_{0^*}$) and 560 °C, 2.3 GPa ($P_1$). The corresponding values of $\beta_0$, $\Delta V_r$ and $V_0$ for reaction 1 are as follows: $\beta_0 = 2.6622\text{E}^{-11}$ (Pa$^{-1}$), $\Delta V_r = 24.03$ (cm$^3$) and $V_0 = 2102.1723$ (cm$^3$), and for reaction 2: $\beta_0 = 2.8598\text{E}^{-11}$ (Pa$^{-1}$), $\Delta V_r = 29.2638$ (cm$^3$), and $V_0 = 2043.2946$ (cm$^3$).

## Data availability

The data used in this study are available from the corresponding author upon request.

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

## Acknowledgements

This work was supported by the Swiss National Foundation grant numbers 26084159, 206021_163991, and 200021-153094. C.L. thanks M. Robyr for assistance during the EPMA data acquisition process. Colormap davos (http://www.fabiocrameri.ch/colourmaps.php) was used for Fig. 6.

## Author contributions

C.L. performed the data acquisition and analysis, made Figs. 1–5 and Supplementary Figs. 1–2, and wrote the first version of the manuscript. L.B. and S.M.S. designed the research project, contributed to the interpretation of the data, and improved the final manuscript version. S.M.S. made the Fig. 6 and S3. G.S. contributed to the SIMS methodology development, the data acquisition, and data reduction processes. T.V. developed and performed the TC/EA analyses on the white mica reference materials.

## Competing interests

The authors declare no competing interests.
