## [Peer Review File · Nature Communications]

Reviewers' comments:

Reviewer #1 (Remarks to the Author):

This is a very provocative paper on "Metamorphic pressure variation in coherent Alpine nappe challenges lithostatic paradigm". Challenging a paradigm is an ambitious goal and for doing so it would need a well-founded manuscript, which, in my opinion the current manuscript is unfortunately not. Documentation of the case is, at least in part, unsatisfactory and overall the manuscript is too speculative and thus too weak for challenging a major paradigm. The authors need to rule out that the Si variations are not a metasomatic overprint; therefore independent P,T estimations are needed (see below). Also, the 3D shape of the whiteschist body needs to be shown and how this 3D shape relates to the numerical model (see below).

There is not enough documentation that the whiteschist represents a fluid pipe. Why is it not a xenolith? In Fig.1c it is not clear whether the photo represents an XZ section of the finite strain ellipsoid or a section parallel or oblique to the main foliation? How does Figure 1c relate to Figure 1b? The reddish whiteschist in Fig.1b looks like apophyses in granite to me. How do those sections relate to the modelling in Figure 4 and compressive stress etc.? One cannot constrain compressive stress in heterogeneously deformed rocks but one could constrain finite strain directions in those rocks. Does the long axis of the modelled elliptical inclusion represent the X axis of the finite strain ellipsoid? To me the 3D aspects of the whole study remains unclear.

The PT estimates are solely based on Si content in white mica of the very aluminous whiteschist. Could the highly varying Si content simply result from metasomatism?

If so, it would perfectly explain that Si content is decreasing away from whiteschist. In other words, Si is added to the granite due to metasomatic processes related to whiteschist formation. The huge variation in Si content would be in line with metasomatism and the gradient in Si content could be explained by a metasomatic halo around the whiteschist. The latter would also be in line with epidote and white mica replacing plagioclase likely related to fluid flow. One could argue that this would be simpler explanation than metamorphic pressure variations. If the authors could rule out metasomatism, then they might have a case. However, for making the study more robust I would recommend to try and estimate P and T independently, for instance by using the average PT method and check if it fits with the pressure results from the Si content data. Taking the highest Si content of the highly varying white mica might be ok, but I do not see a good reason for doing so.

I am sorry to be critical but I think for the authors need to make their case stronger for challenging a major paradigm in metamorphic petrology and tectonics. I do not know whether or not a better documentation of the presented case can be made in a short Nature paper?

Stockholm 27 April 2018

Reviewer #2 (Remarks to the Author):

The manuscript of Luisier et al. addresses the discrepancies between pressure estimates of a whiteschist lens and its host metagranite. The authors then conclude that the apparent high-pressure metamorphism recorded by the whiteschist is due to overpressure in the crust. Deviations from lithostatic pressure have long been speculated and predicted by models, but few field-based evidence has been established.

This manuscript contains two main parts, the first reveals the pressure differences between two lithologic units based on observations and analyses, and the second is a well-defined model for likely explanations. The manuscript as a whole is clearly written and well-structured. I am persuaded that the model is self-consistent and possible, and the authors present convincing evidence of local pressure deviation from lithostatic.

I have some comments that do not challenge the key conclusion of the manuscript, but I would like to point out some technical and conceptual weakness that the authors should address to protect the work from being overinterpreted or misinterpreted.

1) The pressure of host metagranite is constrained by 1) the presence of igneous plagioclase, and 2) Si content in phengite. The first is far more straightforward and convincing than Si-in-phengite barometer, because the latter involves a considerable set of assumptions. The authors elaborate on Si-in-phengite barometer, but a couple of pieces of critical information are missing. For example, the authors fail to explain or demonstrate why the large plagioclase porphyroblasts are igneous relict. Is there any textural or compositional evidence to support their igneous origin? As NatComm papers aim to a broad audience, not all geoscientists can readily correlate plagioclase decomposition to the jadeite-in reaction, or the pressure of this reaction. The authors need to connect the dots so that the readers from another background could quickly get access to the critical message.

The authors rely on Si-in-phengite barometer but do not provide any reference, or the method details. Thus, it is impossible for the readers to assess the quality of the PT work. As far as I am

aware, the Si-in phengite barometer involves equilibrium with quartz, H₂O, and other K- and Al-bearing phases (e.g., Massonne & Schreyer, 1987: K-spar, biotite and quartz), so Si contents are calibrated in specific assemblages. Are these phases present in the peak assemblage? If not, the authors need to justify why this barometer is still applicable, because the phengite barometry is dependent on mineral assemblages (Massonne & Schreyer, 1989). Phase equilibria analysis showed that the Si-in-phengite isopleths are not always pressure-sensitive. In some assemblages, the silica content is dependent on temperature, or bulk Mg# (e.g., Wei & Powell, 2003, CMP). The authors test the sensitivity of pressure on water activity, but do not discuss the activity of silica. If SiO₂ is undersaturated, which is likely in this case because quartz is not present in the assemblage, the calculated pressure is a lower estimate. If the authors adopt a different barometer from another reference, the readers would appreciate a description of the method.

Another concern is the scatter of Si contents in the muscovite. The authors use the highest Si contents to represent the upper limits of pressures, and suggest that lower Si contents represent a failure to reach equilibrium. Then would it be possible that the phengites with highest Si contents did not chemically equilibrate with the matrix either? It bothers me because the authors argue that the absence of jadeite was not due to sluggish kinetics (Line 73), but use reaction kinetics to explain the scattering of Si contents in phengite—note that Si-in-phengite barometers are also dehydration reactions. The arguments need to be logically consistent, or the authors have the burden to explain why the reaction producing phengite is more sluggish than jadeite decomposition. Rigorous statistical approaches are also necessary to show that 1) it is not likely that the highest Si contents were missed in analysis and 2) the apparent increase in Si contents toward whiteschist is statistically significant.

The authors measured OH contents in phengite to demonstrate that water activity was close to unity, by incorrectly applying Raoult's law. Raoult's Law applies to the activity-concentration relationship in a single phase. Unless the authors assume that the (F+Cl)/OH ratios are the same in muscovite and the fluid phase, the measured OH content cannot reflect the water activity. As a matter of fact, the water activity could be lower than unity (a.k.a. no free water) even if the hydroxyl site of muscovite is occupied by OH only.

2) Different barometry methods are applied to different lithologic units (whiteschist and metagranite), so it is not rigorous to compare the pressures without the uncertainties associated with different barometers. I do not doubt that the pressures denoted by the peak assemblages of whiteschist are higher than the metagranite, and I understand that the uncertainties of pseudosection and barometers are not always well defined. However, the authors need to demonstrate, explain, or estimate the uncertainties in their P-T work to readers from other fields. I would like to encourage the authors to add a few sentences and references to justify the ± 0.2 GPa uncertainty (Line 106). In this version, it seems to come out of thin air.

3) In the model, pressure decreases in the host rock at 70-110 m toward the elliptical inclusion. However, the Si contents in phengite increase toward the whiteschist lens. This seems inconsistent to me, but I might miss something. I hope the authors could explain the links between this field observation with the model in the revised manuscript.

Did the whiteschist dyke, between the competent metagranite and soft paragneiss (Fig. 1B), experience similar overpressure? If so, does the same model apply to the whiteschist dyke?

The authors interpret the whiteschist tube as intensively metasomatized granite, which reflects fluid pathway. Could overpressured fluid an alternative mechanism, which could also explain the decrease of pressure in host rock at increasing distance from the fluid channel.

Minor comments:

Line 14. This is a long-standing assumption in petrology. I think it is okay not to cite, or cite a pioneering work by early researchers like Eskola or Korzhinskii.

Line 55-56. Need reference.

Line 63. The presence of titanite and absence of rutile (?) is another piece of evidence that the pressure is lower than 14-15 kbar.

Line 65 & 69. Please add a couple of sentences to explain the igneous origin of plagioclase.

Line 67-68. Please give a couple of references. Is there an independent thermometer to show that the peak temperatures of both

whiteschist and metagranite are same?

Line 71-72. Please explicitly provide the jadeite-producing dehydration reaction and discuss its implication to pressure estimate.

Line 86-87. "Low water activity relates to higher pressures" I could understand what the authors meant, but the sentence is extremely awkward and does not make much sense.

Line 97. See comment (1) above.

Line 110-114. Refs 24 and 25 are studies of pressure gradients recorded by mineral zonation. Ref 23 is a compilation of all HP-UHP terranes around the world, and argues that all HP-UHP terranes are formed by tectonic overpressure. Ref 26 is a petrologic study of a specific outcrop, which is more close to the scope of this study. It confuses the readers when these studies are listed in parallel.

Line 136. The P-T work and the model never address the relative timescales of overpressure, so the “transient” statement in this conclusion paragraph is weak (also in the abstract, line 24). Is this “transient” same as “rapid decompression” in Line 40, or of even shorter timescales?

Line 142-145. This study focuses on a localized feature in Monte Rosa nappe, and the conclusion can only support the overpressure of this whiteschist body, or Monte Rosa nappe at most. The readers don't even know whether the whiteschist dyke, tens of meters from the lens (Fig. 1B), underwent overpressure due to the same mechanism. The authors would have gone too far by stating that the whole Alpine orogen, or Alpine-type collisional orogens, are likely controlled by the same mechanism.

All published overpressure papers (e.g., Refs 22-26) have been highly controversial due to the unintuitive and outlandish conclusions. This study would make a significant contribution to earth science by only making perfectly justifiable interpretations based on the available data, and confirming overpressure of this specific outcrop. Overinterpretation weakens the conclusion, and might lead to unnecessary obstacles to the eventual publication.

Figure 2 and elsewhere, please use standard mineral abbreviations.

Reviewer #3 (Remarks to the Author):

This paper is an interesting contribution to the current vigorous debate regarding the interpretation of pressure in the crust from the point of view of thermodynamic estimates based on mineralogy and lithostatic pressure, compared to pressure estimates from tectonic models of deforming rocks. The crux of the issue is whether there is a simple relation between depth in the crust and pressure (as assumed in the thermodynamic models). This is a very “high-stakes” issue as the lithostatic

paradigm underpins much of metamorphic petrology and determined P,T (& time) paths of rocks in the crust.

The argument presented in this paper is that large pods of rock called whiteschists record a higher pressure than the metagranite rocks in which they are enclosed, and as they formed were metamorphosed at the same time, this is an example of a considerable pressure variation (~0.8GPa) over a relatively small distance. The authors make a strong case that the the pressure differences are not due to later superimposed events affecting only the metagranite.

This interpretation is reinforced by a model of an elliptical softer inclusion within a harder matrix undergoing compression. This model has been invoked in several papers in a more general and abstract way but appears to describe the situation in these rocks.

This is an excellent contribution and will reinforce the current debate that differential pressures can exist in rocks of different strength during deformation. It should be published and I only have minor comments.

1. For readers who are not Alpine geologists it would be helpful to be more explicit about the pre-Alpine petrology of this outcrop. From my understanding of the paper, the protolith was a pre-Alpine plagioclase-bearing granite that contained fluid pathways (tubes) that metasomatized the granite to a different composition. Subsequently, during the Alpine orogeny the whole outcrop was metamorphosed resulting in the metagranite and the tubes become the whiteschist.

If this is not a correct interpretation the authors should make the actual situation clearer.

I would find it helpful to know what the bulk compositions of these two rock types was. Could the 'tubes' have formed during the metamorphism that converted the granite to metagranite, representing fluid pathways formed during the orogeny? i.e that the metasomatism to whiteschist was contemporaneous with the granite metamorphism.

2. In line 6 the authors refer to different metamorphic pressures that are recorded in adjacent rocks of different composition. This compositional difference may be the case here (hence the above question), but it need not be always the case. There are many examples where different pressures are recorded in rocks of closely the same composition.

3. Line 24 and other places. The authors refer to the differential stress as being transient. But the stress has to exist over the duration of the metamorphism which may be xxx years. Can the

authors comment on what they mean by transient, and perhaps also refer to the paper by Dabrowski et al JMG 2015.

4. Of greater concern is the statement in line 65 that partial replacement and pseudomorphism of the igneous plagioclase in the granite indicates local equilibrium. Between which minerals? Surely not between the parent plagioclase and the albite/zoisite replacement. There may be equilibrium between the reaction products and the fluid from which they form, but partial replacement and pseudomorphism are clear indicators of disequilibrium between the parent and product phases (contrary to what is stated by Vernon and others). Albitisation of plagioclase does not indicate that the plagioclase and albite are in equilibrium. The authors should amplify or explain that sentence.

Apart from these points which do not affect the conclusions of this paper, this is a significant contribution and will be referred to when other similar calculated pressure differences are found in deforming rocks in other localities.

5. A final query might be whether equilibrium pseudosections that assume rocks under lithostatic pressure are applicable for pressure calculations for rocks under differential stress. If the authors are challenging the lithostatic paradigm, is there a circular argument here?

Reviewer #1 (Remarks to the Author):

This is a very provocative paper on "Metamorphic pressure variation in coherent Alpine nappe challenges lithostatic paradigm". Challenging a paradigm is an ambitious goal and for doing so it would need a well-founded manuscript, which, in my opinion the current manuscript is unfortunately not. Documentation of the case is, at least in part, unsatisfactory and overall the manuscript is too speculative and thus too weak for challenging a major paradigm. The authors need to rule out that the Si variations are not a metasomatic overprint; therefore independent P,T estimations are needed (see below). Also, the 3D shape of the whiteschist body needs to be shown and how this 3D shape relates to the numerical model (see below).

There is not enough documentation that the whiteschist represents a fluid pipe. Why is it not a xenolith? In Fig. 1c it is not clear whether the photo represents an XZ section of the finite strain ellipsoid or a section parallel or oblique to the main foliation? How does Figure 1c relate to Figure 1b? The reddish whiteschist in Fig. 1b looks like apophyses in granite to me. How do those sections relate to the modelling in Figure 4 and compressive stress etc.? One cannot constrain compressive stress in heterogeneously deformed rocks but one could constrain finite strain directions in those rocks. Does the long axis of the modelled elliptical inclusion represent the X axis of the finite strain ellipsoid? To me the 3D aspects of the whole study remains unclear.

The nature of the protolith of the Monte Rosa whiteschist was addressed by Pawlig (2001) and Pawlig and Baumgartner (2001). Whole rock compositions, stable isotopes as well as detailed field observations support a granite origin of the whiteschist protolith and point toward a late magmatic hydrothermal origin of the metasomatic fluid. Independent radiogenic Sr isotopes confirm a pre-Alpine age of the metasomatic event. Recent unpublished data (in prep.) on the outcrop presented in this study confirm the results of Pawlig (2001) and detailed field observations on this whiteschist outcrop revealed the absence of any structural feature, such as fault or shear zone associated with the whiteschist body. Therefore, the late magmatic hydrothermal fluids must have travelled through the granite by reaction fingering along tube-like 3D structures, generated by hydrothermal fluid circulation cells around the shallow granite intrusion. Geochemical data as well as field observation also rule out a xenolith origin of the whiteschist protolith.

Details about the origin of the whiteschist metasomatism were added in lines 36-48, in order to improve the understanding of the pre-Alpine and Alpine evolution of the whiteschist and metagranite. Axes (x, y, z) were added in Fig. 1B and Fig. 7 to clarify the orientation of the model configuration relatively to the 3D shape of the whiteschist body. The long axis of the ellipse in the model (Fig. 7) represents the tube length in a x-z section of the 3D sketch (Fig. 1B). In Fig. 1C, the outcrop picture is illustrated in the x-y section in the 3D sketch (Fig. 1B) and "outcrop surface" label was added in order to clarify it.

There was no strong deformation associated with the prograde and high-pressure metamorphism in the whiteschist and metagranite, hence it is impossible to constrain the finite instantaneous strain at high-pressure. Also, given the irregular shape and orientation of the tube-like whiteschist body, it is impossible to know the exact orientation of the tube relatively to the compressive force at high-pressure during burial of the nappe. However, the whiteschist - metagranite transition is sub-vertical in the outcrop surface (Fig. 1C) and the structures associated with the nappe formation and exhumation have a sub-horizontal dipping, therefore it is likely that locally, the tube axis was at high angle ($>60^\circ$) from the compressive stress and this resulted in the modeled overpressure.

The PT estimates are solely based on Si content in white mica of the very aluminous whiteschist. Could the highly varying Si content simply result from metasomatism? If so, it would perfectly explain that Si content is decreasing away from whiteschist. In other words, Si is added to the granite due to metasomatic processes related to whiteschist formation. The huge variation in Si content would be in line with metasomatism and the gradient in Si content could be explained by a metasomatic halo around the whiteschist. The latter would also be in line with epidote and white mica replacing plagioclase likely related to fluid flow. One could argue that this would be simpler explanation than metamorphic pressure variations. If the authors could rule out metasomatism, then they might have a case. However, for making the study more robust I would recommend to try and estimate P and T independently, for instance by using the average PT method and check if it fits with the pressure results from the Si content data. Taking the highest Si content of the highly varying white mica might be ok, but I do not see a good reason for doing so.

The P-T estimates of the whiteschist are not based on the Si content of phengite, they are based on thermodynamic calculations that reproduce the correct paragenesis with correct mode and phase composition. The Si in phengite barometry is only used for the metagranite P estimates.

The decrease in Si content in the metagranite away from the whiteschist cannot be explained by metasomatic enrichment in silica, because Si is not added to the rock during metasomatism (shown by ISOCON data in Fig. 5 from Pawlig and Baumgartner, 2001). Additional unpublished data (in prep.) on the whiteschist presented in this study confirm that silica is perfectly immobile in this case, hence neither removed nor added during metasomatism.

I am sorry to be critical but I think for the authors need to make their case stronger for challenging a major paradigm in metamorphic petrology and tectonics. I do not know whether or not a better documentation of the presented case can be made in a short Nature paper?

Stockholm 27 April 2018

Reviewer #2 (Remarks to the Author):

The manuscript of Luisier et al. addresses the discrepancies between pressure estimates of a whiteschist lens and its host metagranite. The authors then conclude that the apparent high-pressure metamorphism recorded by the whiteschist is due to overpressure in the crust. Deviations from lithostatic pressure have long been speculated and predicted by models, but few field-based evidence has been established.

This manuscript contains two main parts, the first reveals the pressure differences between two lithologic units based on observations and analyses, and the second is a well-defined model for likely explanations. The manuscript as a whole is clearly written and well-structured. I am persuaded that the model is self-consistent and possible, and the authors present convincing evidence of local pressure deviation from lithostatic.

I have some comments that do not challenge the key conclusion of the manuscript, but I would like to point out some technical and conceptual weakness that the authors should address to protect the work from being overinterpreted or misinterpreted.

1) The pressure of host metagranite is constrained by 1) the presence of igneous plagioclase, and 2) Si content in phengite. The first is far more straightforward and convincing than Si-in-phengite barometer, because the latter involves a considerable set of assumptions. The authors elaborate on Si-in-phengite barometer, but a couple of pieces of critical information are missing. For example, the authors fail to explain or demonstrate why the large plagioclase porphyroblasts are igneous relict. Is there any textural or compositional evidence to support their igneous origin? As NatComm papers aim to a broad audience, not all geoscientists can readily correlate plagioclase decomposition to the jadeite-in reaction, or the pressure of this reaction. The authors need to connect the dots so that the readers from another background could quickly get access to the critical message.

The igneous nature of the large plagioclase porphyroblasts is attested by the preservation of igneous twinning. Igneous lamellar twinning, which is typical for plagioclase in granites, is observed both in the unreacted portions of the igneous plagioclase, as well as in the albite pseudomorphs. We added two microphotographs in Figure 3 (C and D), showing the plagioclase twins preserved in both the pseudomorphosed regions and in the igneous plagioclase relicts. The fact that igneous twinning is preserved in the albite pseudomorphs is an evidence that albite grew with the same crystallographic orientation as the plagioclase reactant from which it originates. The possibility that jadeite was ever produced and subsequently retrograded to albite can hence be excluded, since albite product after jadeite destabilization would not reproduce the original igneous twinning. Text explanations were added in lines 100-108.

The relationship between plagioclase breakdown, jadeite formation and eclogite facies in metagranite was highlighted in lines 95-98, by adding the reaction and explaining that jadeite is the eclogite-facies marker in metagranite.

The authors rely on Si-in-phengite barometer but do not provide any reference, or the method details. Thus, it is impossible for the readers to assess the quality of the PT work. As far as I am aware, the Si-in phengite barometer involves equilibrium with quartz, H₂O, and other K- and Al-bearing phases (e.g., Massonne & Schreyer, 1987: K-spar, biotite and quartz), so Si contents are calibrated in specific assemblages. Are these phases present in the peak assemblage? If not, the authors need to justify why this barometer is still applicable, because the phengite barometry is dependent on mineral assemblages (Massonne & Schreyer, 1989). Phase equilibria analysis showed that the Si-in-phengite isopleths are not always pressure-sensitive. In some assemblages, the silica content is dependent on temperature, or bulk Mg# (e.g., Wei & Powell, 2003, CMP). The authors test the sensitivity of pressure on water activity, but do not discuss the activity of silica. If SiO₂ is undersaturated, which is likely in this case because quartz is not present in the assemblage, the calculated pressure is a lower estimate. If the authors adopt a different barometer from another reference, the readers would appreciate a description of the method.

Phengites from metagranites selected for Si-in phengite barometry are located in triple junctions involving biotite, K-feldspar and quartz. Hence the buffering assemblage of Massonne and Schreyer (1987) is present at peak pressure. Silica in phengite calculations were done using Massonne and Szpurka (1997) calibration. These references were added in line 116 and details on the calculations were added in the method section.

Another concern is the scatter of Si contents in the muscovite. The authors use the highest Si contents to represent the upper limits of pressures, and suggest that lower Si contents represent a failure to reach equilibrium. Then would it be possible that the phengites with highest Si contents did not chemically equilibrate with the matrix either? It bothers me because the authors argue that the absence of jadeite was not due to sluggish kinetics (Line 73), but use reaction kinetics to explain the scattering of Si contents in phengite—note that Si-in-phengite barometers are also dehydration reactions. The arguments need to be logically consistent, or the authors have the burden to explain why the reaction producing phengite is more sluggish than jadeite decomposition. Rigorous statistical approaches are also necessary to show that 1) it is not likely that the highest Si contents were missed in analysis and 2) the apparent increase in Si contents toward whiteschist is statistically significant.

Between four and eleven buffered areas per metagranite sample were considered for phengite composition measurement, corresponding to a total number of phengite analyses of: metagranite 1: 329, metagranite 2: 155, metagranite 3: 61, metagranite 4: 66 and metagranite 5: 52. In each sample, the maximum Si content in phengite obtained for each analyzed area was reproducible between not all, but several areas. This suggests that the highest Si content is most probably not missed in analysis and hence the increase in Si content in metagranites towards the whiteschist is most probably not a statistical artefact.

Metagranite 1 was used for statistical analysis of the spatial distribution of the Si content in phengite. The phengite domains around biotite consist of several phengite subgrains, intergrown with fine-grained titanite. Several profiles were measured from the most internal position within the phengite rim towards the exterior. A figure illustrating the spatial variability in Si (p.f.u) was added as Fig. 4. The maximum Si content display a "plateau" shape. Therefore, we assume that the plateau reflects a true maximum in pressure and not a post-peak retrograde re-equilibration (that would show a bell-like shape). A similar spread in silica values is recorded in other examples, such as Chopin and Maluski (1980) and Lardeaux et al. (1983) and is discussed in Evans and Patrick (1987). The Si range is interpreted to be a record of the geological evolution of the terranes from high P-T to low P-T conditions. Therefore, we think that these silica contents reflect rather equilibrium at single time in the P-T evolution during prograde, peak and retrograde history, rather than disequilibrium. The way we formulated it in the text seems to be inappropriate and hence we modified that sentence and added a reference in lines 130-132.

The scale of chemical equilibration of phengite between biotite and the matrix is assumed to be large enough for phengite to reach equilibrium during the high-pressure stage with the buffering mineral assemblage. The presence of titanite together with phengite and garnet in the matrix at a large distance from the biotite (Fig. 4) suggest that diffusion was efficient at least between biotite and the matrix. Note also, that the biotites are not fully armored with coronas, as would be expected in a rock with a large amount of disequilibrium

The authors measured OH contents in phengite to demonstrate that water activity was close to unity, by incorrectly applying Raoult's law. Raoult's Law applies to the activity-concentration relationship in a single phase. Unless the authors assume that the (F+Cl)/OH ratios are the same in muscovite and the fluid phase, the measured OH content cannot reflect the water activity. As a matter of fact, the water activity could be lower than unity (a.k.a. no free water) even if the hydroxyl site of muscovite is occupied by OH only.

We agree that Raoult's law only applies to a single phase. We needed it to apply it to define the water activity in phengite. We have added a section outlining the approach and the simplifications used. Even if there is an uncertainty in converting water content to water activity, water activity $\ll 0.5$ would be very unlikely. See section on methods for the derivations.

2) Different barometry methods are applied to different lithologic units (whiteschist and metagranite), so it is not rigorous to compare the pressures without the uncertainties associated with different barometers. I do not doubt that the pressures denoted by the peak assemblages of whiteschist are higher than the metagranite, and I understand that the uncertainties of pseudosection and barometers are not always well defined. However, the authors need to demonstrate, explain, or estimate the uncertainties in their P-T work to readers from other fields. I would like to encourage the authors to add a few sentences and references

to justify the ± 0.2 GPa uncertainty (Line 106). In this version, it seems to come out of thin air.

The uncertainty on pressure estimates is assessed by propagating the error related to the experimental determination of the reaction albite = jadeite and quartz (± 0.05 GPa; Holland, 1980), the uncertainty on the pressure determination related to the silica in phengite experiments (± 0.03 GPa; Massonne and Szpurka, 1997) and the difference in pressure of the modeled invariant point in the MASH system between experiments and two thermodynamic databases (± 0.2 GPa; Chopin and Schreyer, 1983; Berman, 1988; Holland and Powell, 1998, 2002). The error propagation is done as follow and results in a final uncertainty of 0.2 GPa:

$$\text{uncertainty (GPa)} = \sqrt{0.05^2 + 0.03^2 + 0.2^2}$$

Then, a pressure variation of 0.8 ± 0.3 GPa is mentioned and here, the ± 0.3 GPa results from the propagation of the uncertainty on each pressure estimate (pressure in whiteschist and pressure in metagranite), calculated as:

$$\text{Final uncertainty } (\pm 0.3 \text{ GPa}) = \sqrt{0.2^2 + 0.2^2}$$

Details were added in the text in lines 164-170 together with references.

3) In the model, pressure decreases in the host rock at 70-110 m toward the elliptical inclusion. However, the Si contents in phengite increase toward the whiteschist lens. This seems inconsistent to me, but I might miss something. I hope the authors could explain the links between this field observation with the model in the revised manuscript. Did the whiteschist dyke, between the competent metagranite and soft paragneiss (Fig. 1B), experience similar overpressure? If so, does the same model apply to the whiteschist dyke? The authors interpret the whiteschist tube as intensively metasomatized granite, which reflects fluid pathway. Could overpressured fluid an alternative mechanism, which could also explain the decrease of pressure in host rock at increasing distance from the fluid channel.

The whiteschists are NOT a fluid channel, since the metasomatism occurred late intrusive, hydrothermally. But we took this suggestion to allow us to discuss another potential mechanism – fluid pressure due to dehydration, or general changes in volume of a reacting rock, including fluid pressure. We added this part in the paper. See additional explanations below.

The variation in pressure around the inclusion in the model is either positive or negative depending on the location: when looking at the blue field in the host in Figure 6, pressure increases outward, whereas when looking at the yellowish-greenish areas, pressure decreases outward. However, this shows only a 2D view of the pressure distribution in the host and the width of the pressure perturbation is dependent on the length of the pre-defined ellipse. In

our case, the length of the whiteschist tube is unknown and the exact location of the profile sampled relatively to the model is unknown. However, the good point is that we observe a pressure gradient in the host metagranite and the pressure gradient (being positive or negative) is also reproduced in the model. Details were added in text at lines 202-208.

The whiteschist dyke at the contact between the metagranite and paragneiss was strongly deformed during the exhumation of the nappe and therefore it is retrogressed into a greenschist-facies mineral assemblage. Hence, the maximum pressure could not be calculated in this area and comparison with the whiteschist tube could not be made.

The metasomatic alteration of the granite to whiteschist was established by Pawlig and Baumgartner (2001) and by ourselves in a paper in preparation, where we show that the fluids responsible for the metasomatic alteration of the granite are late magmatic hydrothermal fluids, that infiltrated both at the boundary between the granite and paragneiss and also inside the granite along tube-like structures. These results imply that the granite was already altered and hydrated in these tubes before the onset of Alpine metamorphism and therefore whiteschist were fluid saturated during the prograde high-pressure evolution and they experienced dehydration reactions at high pressure. The hypothesis that the fluids released by dehydration at high-pressure would produce overpressure was addressed. We calculated the upper bound of fluid overpressure produced in the whiteschist by assuming that the metagranite is undeformable and hence fluid production happens in a space with constant volume and impermeable boundaries. Fluid overpressure was calculated for the two major dehydration reactions taking place in the whiteschist of Fig. 2B: 1) from 400 °C, 1.5 GPa to 500 °C, 1.5 GPa and 2) from 560 °C, 2.0 GPa to 560 °C, 2.3 GPa. The results indicate a fluid overpressure of 0.43 GPa in reaction 1) and 0.5 GPa in reaction 2), larger than the uncertainty on the pressure estimate.

The fluid-overpressure hypothesis and the calculations were added to the manuscript in lines 216-241 and explained in the method section.

If the metagranite would have been deformable in an elastic or viscous manner then also a relative pressure decrease in the metagranite away from the whiteschist could be possible (see for example the model in Dabrowski et al., J. Met. Geol., 2015). However, a detailed analysis on the potential viscous and elastic deformation in the metagranite is beyond the scope of our model.

Minor comments:

Line 14. This is a long-standing assumption in petrology. I think it is okay not to cite, or cite a pioneering work by early researchers like Eskola or Korzhinskii.

Line 55-56. Need reference.

Line 63. The presence of titanite and absence of rutile (?) is another piece of evidence that the pressure is lower than 14-15 kbar.

Yes, however we chose not to mention titanite as an evidence for pressure lower than 15 kbar because we don't have the exact titanite compositions and depending on the composition, the stability field can reach higher pressure.

Line 65 & 69. Please add a couple of sentences to explain the igneous origin of plagioclase. The presence of igneous twinning is reported (and shown in Fig. 3C-D), that attests the igneous nature of the plagioclase. Textural relationships between the twins and the metamorphic albite growth is discussed in lines 102-108.

Line 67-68. Please give a couple of references. Is there an independent thermometer to show that the peak temperatures of both whiteschist and metagranite are same?

To our knowledge, there is no published independent temperature estimate on the metagranite.

Line 71-72. Please explicitly provide the jadeite-producing dehydration reaction and discuss its implication to pressure estimate.

Line 86-87. "Low water activity relates to higher pressures" I could understand what the authors meant, but the sentence is extremely awkward and does not make much sense.

Line 97. See comment (1) above.

Line 110-114. Refs 24 and 25 are studies of pressure gradients recorded by mineral zonation. Ref 23 is a compilation of all HP-UHP terranes around the world, and argues that all HP-UHP terranes are formed by tectonic overpressure. Ref 26 is a petrologic study of a specific outcrop, which is more close to the scope of this study. It confuses the readers when these studies are listed in parallel.

Line 136. The P-T work and the model never address the relative timescales of overpressure, so the "transient" statement in this conclusion paragraph is weak (also in the abstract, line 24). Is this "transient" same as "rapid decompression" in Line 40, or of even shorter timescales?

The timescale of the high differential stress episode would be shorter than the burial and exhumation cycle. The differential stress builds up under high strain rate, however the lack of substantial deformation in the metagranite at high-pressure implies a short duration of the high strain rate event. Therefore, we suggest that high differential stresses are short-lived (<1Ma) and possibly related to the stage of detachment of the Monte Rosa nappe from the underthrusting continental crust, at maximal burial depth. The differential stress would be

subsequently viscously relaxed during ongoing exhumation of the nappe, as suggested in Dabrowski et al. (2015).

A sentence was added in lines 249-252 to clarify the term "transient".

Line 142-145. This study focuses on a localized feature in Monte Rosa nappe, and the conclusion can only support the overpressure of this whiteschist body, or Monte Rosa nappe at most. The readers don't even know whether the whiteschist dyke, tens of meters from the lens (Fig. 1B), underwent overpressure due to the same mechanism. The authors would have gone too far by stating that the whole Alpine orogen, or Alpine-type collisional orogens, are likely controlled by the same mechanism.

All published overpressure papers (e.g., Refs 22-26) have been highly controversial due to the unintuitive and outlandish conclusions. This study would make a significant contribution to earth science by only making perfectly justifiable interpretations based on the available data, and confirming overpressure of this specific outcrop. Overinterpretation weakens the conclusion, and might lead to unnecessary obstacles to the eventual publication.

In order to avoid overinterpretation, we modified the final sentence.

Figure 2 and elsewhere, please use standard mineral abbreviations.

Reviewer #3 (Remarks to the Author):

This paper is an interesting contribution to the current vigorous debate regarding the interpretation of pressure in the crust from the point of view of thermodynamic estimates based on mineralogy and lithostatic pressure, compared to pressure estimates from tectonic models of deforming rocks. The crux of the issue is whether there is a simple relation between depth in the crust and pressure (as assumed in the thermodynamic models). This is a very "high-stakes" issue as the lithostatic paradigm underpins much of metamorphic petrology and determined P,T (& time) paths of rocks in the crust.

The argument presented in this paper is that large pods of rock called whiteschists record a higher pressure than the metagranite rocks in which they are enclosed, and as they formed were metamorphosed at the same time, this is an example of a considerable pressure variation (~0.8GPa) over a relatively small distance. The authors make a strong case that the the pressure differences are not due to later superimposed events affecting only the metagranite.

This interpretation is reinforced by a model of an elliptical softer inclusion within a harder matrix undergoing compression. This model has been invoked in several papers in a more general and abstract way but appears to describe the situation in these rocks.

This is an excellent contribution and will reinforce the current debate that differential pressures can exist in rocks of different strength during deformation. It should be published

and I only have minor comments.

1. For readers who are not Alpine geologists it would be helpful to be more explicit about the pre-Alpine petrology of this outcrop. From my understanding of the paper, the protolith was a pre-Alpine plagioclase-bearing granite that contained fluid pathways (tubes) that metasomatized the granite to a different composition. Subsequently, during the Alpine orogeny the whole outcrop was metamorphosed resulting in the metagranite and the tubes become the whiteschist.

If this is not a correct interpretation the authors should make the actual situation clearer. I would find it helpful to know what the bulk compositions of these two rock types was. Could the 'tubes' have formed during the metamorphism that converted the granite to metagranite, representing fluid pathways formed during the orogeny? i.e that the metasomatism to whiteschist was contemporaneous with the granite metamorphism.

Details about the origin of whiteschist were added in few sentences in lines 37-48. The origin of the Monte Rosa whiteschist has been well constrained by Pawlig and Baumgartner (2001) and Pawlig (2001) and their pre-Alpine origin is confirmed by recent unpublished data, where we exclude the possibility that they formed during high-pressure Alpine metamorphism. The late magmatic hydrothermal nature of the metasomatic fluids was constrained based on ISOCON study as well as stable isotopes on whiteschist and associated carbonates. The pre-Alpine age of metasomatism is independently confirmed by radiogenic Sr isotopes data on whiteschist (Pawlig, 2001). Hence the chemistry of whiteschist was established before the onset of Alpine metamorphism and the whiteschist high-pressure mineralogy was acquired under closed-system conditions during the high-pressure Alpine event, along with the high-pressure metagranite mineralogy.

The bulk rock composition of the whiteschist and metagranite are indicated in the method section.

2. In line 6 the authors refer to different metamorphic pressures that are recorded in adjacent rocks of different composition. This compositional difference may be the case here (hence the above question), but it need not be always the case. There are many examples where different pressures are recorded in rocks of closely the same composition.

We slightly modified that sentence

3. Line 24 and other places. The authors refer to the differential stress as being transient. But the stress has to exist over the duration of the metamorphism which may be xxx years. Can the authors comment on what they mean by transient, and perhaps also refer to the paper by Dabrowski et al JMG 2015.

The timescale of the high differential stress episode would be shorter than the burial and exhumation cycle. The differential stress builds up under presumably high strain rate, however the lack of substantial deformation in the metagranite at high-pressure implies a

short duration of the high strain rate event. Therefore, we suggest that high differential stresses are short-lived (<1Ma) and possibly related to the stage of detachment of the Monte Rosa nappe from the subducting continental crust, at approximately maximal burial depth. The differential stress would be subsequently viscously relaxed during ongoing exhumation of the nappe, as suggested in Dabrowski et al. (2015).

A sentence was added in line 249-252 to clarify the term "transient".

4. Of greater concern is the statement in line 65 that partial replacement and pseudomorphism of the igneous plagioclase in the granite indicates local equilibrium. Between which minerals? Surely not between the parent plagioclase and the albite/zoisite replacement. There may be equilibrium between the reaction products and the fluid from which they form, but partial replacement and pseudomorphism are clear indicators of disequilibrium between the parent and product phases (contrary to what is stated by Vernon and others). Albitisation of plagioclase does not indicate that the plagioclase and albite are in equilibrium. The authors should amplify or explain that sentence.

We use the term "local equilibrium" in order to define the scale over which chemical exchange is effective during reaction in the metagranite. In this view, pseudomorphs and partial replacement microstructures indicate that equilibration is restricted in size and that the reaction width is not at the scale of the bulk rock but rather between touching minerals. The albite and zoisite pseudomorphs have together the same bulk compositions as the initial plagioclase. Except water, no chemical exchange with the matrix is needed.

Apart from these points which do not affect the conclusions of this paper, this is a significant contribution and will be referred to when other similar calculated pressure differences are found in deforming rocks in other localities.

5. A final query might be whether equilibrium pseudosections that assume rocks under lithostatic pressure are applicable for pressure calculations for rocks under differential stress. If the authors are challenging the lithostatic paradigm, is there a circular argument here?

Equilibrium pseudosections assume only a pressure with negligible differential stress and not a lithostatic pressure. Pressure in equilibrium pseudosections is defined as $P = (\partial G/\partial V)_S$ and not as $P = \rho gz$. The pressure recorded by mineral assemblages at equilibrium in metamorphic rocks is a result of the local stress state at a given time. In our case, the whiteschist and metagranite are submitted to heterogeneous stress conditions, therefore they record different peak pressures in order to fulfill force balance. However, taken independently, their respective peak mineral paragenesis can be modelled by equilibrium pseudosections as long as pressure in the considered domain is uniform. Moulas et al. 2018 (accepted online in J. Met. Geol.) show that the mean stress in a weak inclusion with minor differential stress is a good approximation to the thermodynamic pressure used for mineral phase equilibrium

calculations. Hence, the mean stress in the weak whiteschist, which can deviate significantly from the lithostatic pressure, can be used for pseudosection modelling.

Reviewers' comments:

Reviewer #1 (Remarks to the Author):

Second review of Luisier et al.

(I made some remarks directly in the rebuttal and then added my second review below their rebuttal. I tried to highlight my words somehow but this is hard in this on-line system as colours, different fonts etc. are not supported by it).

Reviewer #1 (Remarks to the Author): This is a very provocative paper on "Metamorphic pressure variation in coherent Alpine nappe challenges lithostatic paradigm". Challenging a paradigm is an ambitious goal and for doing so it would need a well-founded manuscript, which, in my opinion the current manuscript is unfortunately not. Documentation of the case is, at least in part, unsatisfactory and overall the manuscript is too speculative and thus too weak for challenging a major paradigm. The authors need to rule out that the Si variations are not a metasomatic overprint; therefore independent P,T estimations are needed (see below). Also, the 3D shape of the whiteschist body needs to be shown and how this 3D shape relates to the numerical model (see below). There is not enough documentation that the whiteschist represents a fluid pipe. Why is it not a xenolith? In Fig.1c it is not clear whether the photo represents an XZ section of the finite strain ellipsoid or a section parallel or oblique to the main foliation? How does Figure 1c relate to Figure 1b? The reddish whiteschist in Fig.1b looks like apophyses in granite to me. How do those sections relate to the modelling in Figure 4 and compressive stress etc.? One cannot constrain compressive stress in heterogeneously deformed rocks but one could constrain finite strain directions in those rocks. Does the long axis of the modelled elliptical inclusion represent the X axis of the finite strain ellipsoid? To me the 3D aspects of the whole study remains unclear.

The nature of the protolith of the Monte Rosa whiteschist was addressed by Pawlig (2001) and Pawlig and Baumgartner (2001). Whole rock compositions, stable isotopes as well as detailed field observations support a granite origin of the whiteschist protolith and point

(".. point..." is quite vague)

toward a late magmatic hydrothermal origin of the metasomatic fluid. Independent radiogenic Sr isotopes confirm a pre-Alpine age of the metasomatic event. Recent unpublished data (in prep.) on the outcrop presented in this study confirm the results of Pawlig (2001) and detailed field

observations on this whiteschist outcrop revealed the absence of any structural feature, such as fault or shear zone associated with the whiteschist body.

(As the author's say, this is unpublished data and do not add information for the reader.)

Therefore, the late magmatic hydrothermal fluids must have travelled through the granite by reaction fingering along tube-like 3D structures, generated by hydrothermal fluid circulation cells around the shallow granite intrusion. Geochemical data as well as field observation also rule out a xenolith origin of the whiteschist protolith.

(Apparently unpublished data as well?)

Details about the origin of the whiteschist metasomatism were added in lines 36-48, in order to improve the understanding of the pre-Alpine and Alpine evolution of the whiteschist and metagranite. Axes (x, y, z) were added in Fig. 1B and Fig. 7 to clarify the orientation of the model configuration relatively to the 3D shape of the whiteschist body. The long axis of the ellipse in the model (Fig. 7) represents the tube length in a x-z section of the 3D sketch (Fig. 1B). In Fig. 1C, the outcrop picture is illustrated in the x-y section in the 3D sketch (Fig. 1B) and "outcrop surface" label was added in order to clarify it.

(I do not understand this. In the new manuscript the authors say that the whiteschist is undeformed (see also immediately below here) and that the surrounding metagranite shows greenschist-facies deformation. To which structures do X, Y, Z relate to? I assume that X, Y, Z represent the principal finite-strain axes?)

There was no strong deformation associated with the prograde and high-pressure metamorphism in the whiteschist and metagranite, hence it is impossible to constrain the finite instantaneous strain at high-pressure.

(See my comment just above, this (i.e. constraining finite strain) is exactly what you did by reconstructing X, Y, Z (which I think refer to finite-strain axes) but you say it is impossible to do. More importantly, you do not know at all whether or not the granite was deformed at high-P metamorphism.)

Also, given the irregular shape and orientation of the tube-like whiteschist body, it is impossible to know the exact orientation of the tube relatively to the compressive force at high-pressure during burial of the nappe. However, the whiteschist - metagranite transition is sub-vertical in the outcrop surface (Fig. 1C) and the structures associated with the nappe formation and exhumation have a sub-horizontal dipping, therefore it is likely that locally, the tube axis was at high angle ($>60^\circ$) from the compressive stress and this resulted in the modeled overpressure.

(To me it appears likely, that the stress axes used in the model are unconstrained with respect to the finite-strain axes hypothesised for the field example.)

The PT estimates are solely based on Si content in white mica of the very aluminous whiteschist. Could the highly varying Si content simply result from metasomatism? If so, it would perfectly explain that Si content is decreasing away from whiteschist. In other words, Si is added to the granite due to metasomatic processes related to whiteschist formation. The huge variation in Si content would be in line with metasomatism and the gradient in Si content could be explained by a metasomatic halo around the whiteschist. The latter would also be in line with epidote and white mica replacing plagioclase likely related to fluid flow. One could argue that this would be simpler explanation than metamorphic pressure variations. If the authors could rule out metasomatism, then they might have a case. However, for making the study more robust I would recommend to try and estimate P and T independently, for instance by using the average PT method and check if it fits with the pressure results from the Si content data. Taking the highest Si content of the highly varying white mica might be ok, but I do not see a good reason for doing so.

The P-T estimates of the whiteschist are not based on the Si content of phengite, they are based on thermodynamic calculations that reproduce the correct paragenesis with correct mode and phase composition. The Si in phengite barometry is only used for the metagranite P estimates. See below, why are the different rock types treated differently?

The decrease in Si content in the metagranite away from the whiteschist cannot be explained by metasomatic enrichment in silica, because Si is not added to the rock during metasomatism (shown by ISOCON data in Fig. 5 from Pawlig and Baumgartner, 2001). Additional unpublished data (in prep.) on the whiteschist presented in this study confirm that silica is perfectly immobile in this case, hence neither removed nor added during metasomatism. (see comments above: again unpublished data...)

I am sorry to be critical but I think for the authors need to make their case stronger for challenging a major paradigm in metamorphic petrology and tectonics. I do not know whether or not a better documentation of the presented case can be made in a short Nature paper?

Stockholm 27 April 2018

New, second review

P-T estimates

To be honest, I remain completely unconvinced about the authors reply on the P-T estimates. In my previous review I asked them to use an independent barometer to confirm their findings. They could easily have done so, but have chosen not to. Their response refers to the whiteschist, not to the P-gradient (or alteration halo). I suggest advising them to use the same P-T approach for the whiteschist and the halo. Then there would be a convincing case.

I am not sure that the amount of silica needed to change the phengite content of mica would show up on an isochron diagram and without seeing the isochron diagram, I cannot assess if it could be a dilution affect (I could not access the paper to which they refer which is from the Swiss Journal of Geosciences).

Deformation of whiteschist/metagranite and numerical model

The author's claim that the whiteschists are undeformed (l.46/47) (which makes the use of the term "schist" quite awkward from a Structural Geology point of view), and that the "Metagranites closer than 80 meters to the whiteschist have not been analysed, since their high-pressure assemblage was strongly altered during a late greenschist-facies retrogression associated with deformation", meaning that the metagranite, at least close to the undeformed whiteschist is (strongly) deformed. How based on that can one assume "a simple two-dimensional analytical model in which the whiteschist is mimicked by an ellipse surrounded by a mechanically more competent material representing the metagranite"? To me it sounds as if the whiteschist is mechanically more competent and thus undeformed. Therefore saying in l.199-201: "Significant mechanical strength of the metagranite can be justified because field observations show that large volumes of the metagranite have remained essentially undeformed", and in l.210/211: "Therefore, the pressure gradient determined in the host metagranite could well reflect heterogeneous pressure distribution in the strong host" does not hold true, at least not close to the studied undeformed whiteschist on which the entire story hinges. Since the metagranite was infiltrated by metamorphic fluids and affected by associated deformation during greenschist-facies metamorphism, one cannot assume that the granite was strong, impermeable and undeformed during high-pressure metamorphism (l. 223/224). One simply does not know, at least not for the studied example. To me it appears very likely that the metagranite close to the whiteschist suffered deformation during the high-pressure stage, therefore was anisotropic allowing later fluid infiltration and deformation under greenschist-facies deformation, which then erased the earlier high-pressure tectonometamorphic features.

Whiteschist sheared in or not?

I realise that the author's make the point in l.35-36 that: "The whiteschists are not xenoliths or pieces of the subduction channel that could have been incorporated later by tectonic mixing (mélange) into the granite during the Alpine orogeny" and have not been tectonically incorporated into the granite. They argue that this is because: "The absence of shear zones and faults related to the whiteschist outcrop imply tube-like chemical alteration structures (Fig. 1B) and the lack of any high-pressure foliation associated in the whiteschist" (l. 45-47). However, if the whiteschist would have been sheared into the surrounding metagranite then one would start looking for shear zones in the metagranite, not in the whiteschist. Judging from Fig. 1, the metagranite appears definitely deformed. Saying that: "...were generated from metasomatic alteration of the granite by late magmatic hydrothermal fluids related to the cooling of the granite intrusion long before the onset of Alpine orogenesis" is fine but does not exclude that they were sheared into the metagranite in the Alpine subduction zone.

The author's might want to read back to Gerya et al. TECTONICS, VOL. 21, NO. 6, 1056, doi:10.1029/2002TC001406, 2002, who discuss that:

"It may be suspected that the internal structure of orogenic belts with exposed high-pressure and ultrahigh-pressure metamorphic units, commonly referred to as nappes, with a thickness on the order of 10² to 10³ m and a horizontal width of exposure on the order of some 10³ to 10⁴ m, which feature different maximum P and T conditions, and contrasting P -T histories, may in fact partly represent material that has been extruded from a widening subduction channel to form an orogenic root beneath the forearc during ongoing subduction."

And also: "An array of diverse P-T paths rather than a single P-T trajectory is expected to be characteristic for subduction related metamorphic complexes. In fact, this appears to be what is commonly observed in orogenic belts. The characteristic size and shape of the units with an individual history depend on the rheology of the material in the subduction channel. Our simulations indicate that lower viscosities result in enhanced disintegration and mingling, and hence smaller characteristic length scales for coherent units and a marked contrasts between adjacent slices, a structure commonly termed melange, while higher viscosities favor the formation of extensive coherent slices, possibly representing the type of structures commonly referred to as nappes in high-pressure metamorphic terranes."

It all hinges on whether the whiteschist they studied is in-situ or not. Because this is so critical for the argument in this manuscript I would think it may be a good idea to prove that more rigorously.

Minor comments

L.67-69: "Obviously, the lithostatic pressure assumption is crucial to all these arguments and understanding of metamorphic pressure variations recorded in the Monte Rosa nappe". This is worded much too strongly as the higher-pressure units could have been sheared into the

surrounding rocks in the subduction channel (see for instance Gerya et al., 2002, Tectonics, reference is given below). See above.

L.135: "a plateau with decreasing Si-content away from the plateau" is pretty awkward wording.

L.54-56: the sentence "orogenic wedge model¹²... explains nappe formation and stacking by basal accretion and nappe exhumation by ongoing stacking, subsequent backfolding and erosion" does not appear plausible to me as in this scenario erosion is assumed to be the sole exhumation mechanism. Dozens of tectonic studies have convincingly shown that this is not the case in this part of the Alps. Next sentence that buoyancy did not play a role. This might be true but there needs to be a reference!! Most people (not me!) worrying on exhumation of (U)HP rocks strongly believe that buoyancy is important.

L.63-64: "...subduction channel model¹³. This model explains nappes as buoyancy-driven plumes rising along a subduction channel from depths >80 km^{13,14}". This does unfortunately not make sense either as nappes are not plumes. This is absolutely wrong terminology.

L.75: Not sure whether "retrogression" is the proper term here as you apparently think the rocks were isothermally decompressed and did not cool during initial exhumation. Same for "retrograde" in l.105.

L.99: Statements like "Interestingly, jadeite has never been reported in the literature..." need to be backed up by references (as you refer to "literature").

L.196/197: I think it would help a general audience if this was explained a bit more "mean stress of 1.4 GPa (centre of Mohr stress circle) and a differential stress of 1.4 GPa (diameter of Mohr stress circle)." The mean stress is $(\sigma_1 + \sigma_3)/2$ and the differential stress is simply $\sigma_1 - \sigma_3$. I did not do a full calculation but I think it means that a σ_1 of 2.1 GPa and a σ_3 of 0.7 GPa has been assumed. Please say how you arrived at those values. Why have they been chosen? What is the basis for these numbers?

L.197/198: "a compression direction of 75° with respect to the inclusion's long axis". This angle comes out of the blue. Why is such an angle chosen? This needs some explanation. (see my general comment above).

L.202/201: Saying that “field observations show that large volumes of the metagranite have remained essentially undeformed” does not help. Only important is the deformation of the metagranite at the contact with the studied whiteschist, and this metagranite appears deformed (Fig. 1), as you explicitly say in your text.

L.250/251: “Such burial depths agree with field-based structural restorations.” Again one of those many statements that would need references. This is a recurring problem in the manuscript. There is simply incomplete documentation. I strongly assume that the restoration cannot estimate burial depths of nappes and that the latter have either been inferred or taken from field studies. If so, this is a circular argument.

L.255/256. Same here, “exhumation cycle of the Monte Rosa nappe and was supposedly short-lived (<< 1 Ma).” This statement is absolutely not in line with what has been said in l.54-56. If one exhumes high-P rocks that extremely fast and is inferring erosion to be the main agent, then one needs erosion rates of >>4cm/yr (current average erosion rates in the Alps are <1mm/yr).

Stockholm 2 September 2018

Reviewer #2 (Remarks to the Author):

I am satisfied that the authors have adequately addressed the most of the reviewers' queries. A considerable amount of the reviewers' comments were misinterpretations and confusions due to the presentation of the previous version, and I believe the authors endeavored to convey their thoughts in the better-described Methods and more elaborate Discussion sections. Although the authors do not give quantitative answers to all the questions, these were indeed beyond the scope of this study and did not challenge the main conclusion of this study. I think the paper will be an excellent contribution to the metamorphic and Alpine communities, and the exciting story will stimulate further inquiries and studies. Nonetheless, I still have a few concerns that the authors might want to address before the manuscript is suitable for publication in a Nature journal.

I agree with the authors that the water activity in the metagranite is unlikely much smaller than unity, and the pressure estimate is not very sensitive to $a_{\text{H}_2\text{O}}$. However, the derivation, as the

authors present it (Lines 316-353), creates confusion because we usually define the activity and μ_0 of the $\text{KAl}_3\text{Si}_3\text{O}_{10}(\text{OH})_2$ endmember instead of H_2O in muscovite. The physical meanings of “ $\mu_0_{\text{H}_2\text{O}}^{\text{phen}}$ ” and “activity of water in phengite” are vague. Also note that $X_{\text{H}_2\text{O}}^{\text{phen}}$ in the authors’ formalism is not $\text{OH}/2$ (Line 283), but is $(\text{OH}/2)/(1 + \text{OH}/2 + F + \dots)$ in which “1” stands for the hypothetical $\text{KAl}_3\text{Si}_3\text{O}_{11}$ endmember. Even if we assume that Equation 3 is qualitatively correct, $a_{\text{H}_2\text{O}}^{\text{fl}}$ is not necessarily 1 when “ $a_{\text{H}_2\text{O}}^{\text{phen}}$ ” (which is approximated by $X_{\text{H}_2\text{O}}^{\text{phen}}$) is 1 (Line 342). In addition, their derivation also assumes fluid presence, which, the authors also agree, is possibly not the case (Line 153).

That said, I do not question the pressure estimate for metagranite. I believe Fig. 5 alone (without the grey highlighting) would suffice the purpose. The authors could remove the $a_{\text{H}_2\text{O}}$ derivations from the Method section, and replace $a_{\text{H}_2\text{O}}$ with X_{OH} in Figure 2. Too many assumptions made in the $a_{\text{H}_2\text{O}}$ estimate distract the readers and weaken the results.

Minor comments:

Lines 83-86. This sentence of analytical method should be removed from the Result section.

Line 90. The results of metagranite could be presented in a new paragraph

Line 94. The “which ...” clause should be reworded, because “which” can also be interpreted as “the absence of jadeite”.

Line 114. larger -> higher?

Line 133. A few words seem missing: the proposition of WHAT?

Line 142. pressure increases ... -> the calculated pressure increases ...

Line 151-152. “The activities are considerably lower than in the whiteschists.” This sentence contradicts Line 157 “Water activities close to unity ...”

Line 156. Pressure gradient should be GPa/m, or the authors could replace “pressure gradient”.

Line 168. “whiteschists” should be “metagranites”?

Line 178. “Ref 23” should be Ref 28?

Line 249. What does “more moderate” mean? “More moderate” than what?

Line 309. Please define all the variables. What do subscripts “1”, “0” and “*” mean?

Reviewer #1 (Remarks to the Author):

Second review of Luisier et al.

(I made some remarks directly in the rebuttal and then added my second review below their rebuttal. I tried to highlight my words somehow but this is hard in this on-line system as colours, different fonts etc. are not supported by it).

Reviewer #1 (Remarks to the Author):

This is a very provocative paper on "Metamorphic pressure variation in coherent Alpine nappe challenges lithostatic paradigm". Challenging a paradigm is an ambitious goal and for doing so it would need a well-founded manuscript, which, in my opinion the current manuscript is unfortunately not. Documentation of the case is, at least in part, unsatisfactory and overall the manuscript is too speculative and thus too weak for challenging a major paradigm. The authors need to rule out that the Si variations are not a metasomatic overprint; therefore independent P,T estimations are needed (see below). Also, the 3D shape of the whiteschist body needs to be shown and how this 3D shape relates to the numerical model (see below).

It is fair to be extremely critical towards studies that challenge an existing paradigm, but we find that most of the critical comments of the reviewer are based on misunderstanding and misinterpreting of the data and arguments. We have added material in the appendix, and really taking care to try to identify the sources of misunderstanding. In fact, two of his main critical points, namely that (1) the Si variation is a metasomatic overprint, resetting a high pressure signature and that (2) the whiteschist body is a xenolith, are easily dispelled by either geochemical, petrologic, or field observations. They are not supported at all by our observations, data and previous work, published in an international journal. We will explain our arguments in detail below and we have also modified the manuscript to explain our data and results clearer.

There is not enough documentation that the whiteschist represents a fluid pipe. Why is it not a xenolith? In Fig.1c it is not clear whether the photo represents an XZ section of the finite strain ellipsoid or a section parallel or oblique to the main foliation? How does Figure 1c relate to Figure 1b? The reddish whiteschist in Fig.1b looks like apophyses in granite to me. How do those sections relate to the modelling in Figure 4 and compressive stress etc.? One cannot constrain compressive stress in heterogeneously deformed rocks but one could constrain finite strain directions in those rocks. Does the long axis of the modelled elliptical inclusion represent the X axis of the finite strain ellipsoid? To me the 3D aspects of the whole study remains unclear.

The observation of pre-Alpine dykes that crosscut the metagranite and the whiteschist (indicated in our Fig. 1C) shows that the body (which then was transformed to a whiteschist) was there before cessation of magmatism. No magmatic events are known in the area of Alpine time (in fact all magmatism is associated with the ending of the Hercynian orogenic cycle). There is a strong similarity in chemical composition between metagranite and whiteschist (published in detail in Pawlig and Baumgartner, Schweiz. Mineral. Petrogr. Mitt., 2001) documenting a progressive alteration between these rocks. Finally, the absence of significant peak-pressure shear zones surrounding the body is clear in the field. All this clearly shows that it is not an Alpine introduced xenolith, which has been "tectonically

mixed” into the metagranite from a depth of >20 km deeper than the metagranite. To further show proof of the granite origin of the whiteschist body we have included an additional figure in the appendix, showing the transition zone between the metagranite and the whiteschist in detail, with igneous quartz continuing into the whiteschist. There is no doubt, the body formed from a granite protolith.

The nature of the protolith of the Monte Rosa whiteschist was addressed by Pawlig (2001) and Pawlig and Baumgartner (2001). Whole rock compositions, stable isotopes as well as detailed field observations support a granite origin of the whiteschist protolith and point

(“.. point...” is quite vague)

Actually, these published data demonstrate that the metasomatic fluid is a late magmatic (hence pre-Alpine) hydrothermal fluid. For us, and for the reviewers and editor who published the article by Pawlig and Baumgartner (2001), the “point” towards a granite origin of the whiteschist protolith is very strong.

toward a late magmatic hydrothermal origin of the metasomatic fluid. Independent radiogenic Sr isotopes confirm a pre-Alpine age of the metasomatic event. Recent unpublished data (in prep.) on the outcrop presented in this study confirm the results of Pawlig (2001) and detailed field observations on this whiteschist outcrop revealed the absence of any structural feature, such as fault or shear zone associated with the whiteschist body.

(As the author’s say, this is unpublished data and do not add information for the reader.)

See below

Therefore, the late magmatic hydrothermal fluids must have travelled through the granite by reaction fingering along tube-like 3D structures, generated by hydrothermal fluid circulation cells around the shallow granite intrusion. Geochemical data as well as field observation also rule out a xenolith origin of the whiteschist protolith.

(Apparently unpublished data as well?)

Additional data are part of the PhD thesis of Pawlig mentioned above. Furthermore, the xenolith origin of the whiteschist can be ruled out by the arguments and observations given above (i.e. crosscutting dykes, similarity in chemical composition and absence of large-displacement shear zones). A more detailed description of the outcrop was made in lines 47-61.

Details about the origin of the whiteschist metasomatism were added in lines 36-48, in order to improve the understanding of the pre-Alpine and Alpine evolution of the whiteschist and metagranite. Axes (x, y, z) were added in Fig. 1B and Fig. 7 to clarify the orientation of the model configuration relatively to the 3D shape of the whiteschist body. The long axis of the ellipse in the model (Fig. 7) represents the tube length in a x-z section of the 3D sketch (Fig. 1B). In Fig. 1C, the outcrop picture is illustrated in the x-y section in the 3D sketch (Fig. 1B) and "outcrop surface" label was added in order to

clarify it.

(I do not understand this. In the new manuscript the authors say that the whiteschist is undeformed (see also immediately below here) and that the surrounding metagranite shows greenschist-facies deformation. To which structures do X, Y, Z relate to? I assume that X, Y, Z represent the principal finite-strain axes?)

The X-Y-Z axes of figure 1B refer to the three spatial directions (NW-NE and vertical direction) and not to the finite strain axes. We never mentioned finite strain axes in our manuscript and are, hence, surprised that the reviewer assumes that the X-Y-Z coordinates indicate finite strain axes. To make the text clearer, we replaced the X-Y-Z axes of fig. 1 by NE-NW-vertical coordinates and specified these coordinates in the figure caption of fig. 6.

We discuss the deformation of the metagranite below, in the second review part, however, contrary to the statement of the reviewer, we mentioned in line 47-48 of the manuscript (before revision) that the whiteschist is locally deformed: "...local deformation in the whiteschist pods post-dates Alpine peak pressure.". However, we rephrased that sentence to make it clear, line 61-63.

There was no strong deformation associated with the prograde and high-pressure metamorphism in the whiteschist and metagranite, hence it is impossible to constrain the finite instantaneous strain at high-pressure.

(See my comment just above, this (i.e. constraining finite strain) is exactly what you did by reconstructing X, Y, Z (which I think refer to finite-strain axes) but you say it is impossible to do. More importantly, you do not know at all whether or not the granite was deformed at high-P metamorphism.)

See the discussion about deformation below.

Also, given the irregular shape and orientation of the tube-like whiteschist body, it is impossible to know the exact orientation of the tube relatively to the compressive force at high-pressure during burial of the nappe. However, the whiteschist - metagranite transition is sub-vertical in the outcrop surface (Fig. 1C) and the structures associated with the nappe formation and exhumation have a sub-horizontal dipping, therefore it is likely that locally, the tube axis was at high angle ($>60^\circ$) from the compressive stress and this resulted in the modeled overpressure.

(To me it appears likely, that the stress axes used in the model are unconstrained with respect to the finite-strain axes hypothesised for the field example.)

In fig. 6 we provide a systematic study showing that there is a range of orientations (ca 70 to 90 degrees) and a range of aspect ratios (larger than 4) that result in a pressure of around 2 GPa in the inclusion. The main aim of the presented simple mechanical inclusion model is to show that pressure differences of ca. 0.8 GPa are mechanically feasible. Of course, the natural situation is much more complicated but the aim of simple models is to simplify complex natural situations and to show mechanical feasibilities of the observations, to at least within an order of magnitude. Therefore, our model shows that our interpretations of pressure differences is mechanically feasible for the considered geological situation. We do not argue that the presented inclusion model is adequate to explain all the aspects of the pressure difference, but that this model shows a feasible process, which could have generated the

pressure difference. We also present a second simple model considering reaction-induced volume changes and associated stresses as a potential explanation for the pressure difference.

The PT estimates are solely based on Si content in white mica of the very aluminous whiteschist. Could the highly varying Si content simply result from metasomatism? If so, it would perfectly explain that Si content is decreasing away from whiteschist. In other words, Si is added to the granite due to metasomatic processes related to whiteschist formation. The huge variation in Si content would be in line with metasomatism and the gradient in Si content could be explained by a metasomatic halo around the whiteschist. The latter would also be in line with epidote and white mica replacing plagioclase likely related to fluid flow. One could argue that this would be simpler explanation than metamorphic pressure variations. If the authors could rule out metasomatism, then they might have a case. However, for making the study more robust I would recommend to try and estimate P and T independently, for instance by using the average PT method and check if it fits with the pressure results from the Si content data. Taking the highest Si content of the highly varying white mica might be ok, but I do not see a good reason for doing so.

The P-T estimates of the whiteschist are not based on the Si content of phengite, they are based on thermodynamic calculations that reproduce the correct paragenesis with correct mode and phase composition. The Si in phengite barometry is only used for the metagranite P estimates. See below, why are the different rock types treated differently? The decrease in Si content in the metagranite away from the whiteschist cannot be explained by metasomatic enrichment in silica, because Si is not added to the rock during metasomatism (shown by ISOCON data in Fig. 5 from Pawlig and Baumgartner, 2001). Additional unpublished data (in prep.) on the whiteschist presented in this study confirm that silica is perfectly immobile in this case, hence neither removed nor added during metasomatism. (see comments above: again unpublished data...)

We respond to the critics concerning different P-T estimates in detail below.

I am sorry to be critical but I think for the authors need to make their case stronger for challenging a major paradigm in metamorphic petrology and tectonics. I do not know whether or not a better documentation of the presented case can be made in a short Nature paper?

Stockholm 27 April 2018

New, second review

P-T estimates

To be honest, I remain completely unconvinced about the authors reply on the P-T estimates. In my previous review I asked them to use an independent barometer to confirm their findings. They could easily have done so, but have chosen not to. Their response refers to the whiteschist, not to the P-gradient (or alteration halo). I suggest advising them to use the same P-T approach for the whiteschist and the halo. Then there would be a convincing case.

I am not sure that the amount of silica needed to change the phengite content of mica would show up on an isochron diagram and without seeing the isochron diagram, I cannot

assess if it could be a dilution affect (I could not access the paper to which they refer which is from the Swiss Journal of Geosciences).

Both the whiteschist and metagranite were modeled using the same Domino software and the same Berman's database to provide consistent results. Also, the same solution models were used (Massonne and Szpurka 1997 for white mica) for consistency. To make the text clearer, we added details in the method section in lines 363-367.

There is indeed a major difference, thermodynamically speaking, between phengite content in the whiteschist and the phengite content in the granite. In the whiteschist, phengite content is determined due to whole rock chemistry, while the phengite content in the granite is buffered by the mineral assemblage. This means that the phengite barometry in the granite is much more robust, than it is in the whiteschist. As a consequence, we relied on the phase assemblage domain (field) for the whiteschists, and on the phengite isopleths in the granite. However, for curiosity, we did calculate the silica in phengite isopleths for the whiteschist and the maximum silica content measured in phengite from our sample agree with the modelled silica content of phengite in the modelled peak field assemblage.

Regarding the silica content of phengite and the isocon, yes, this would appear. Silica is one of the most abundant components in granites, so in order to change the silica content in mica only by adding it to the rock, a sufficiently large amount would be required so that it would show up on an isocon. The mentioned article of Pawlig and Baumgartner 2001 (which is actually from the Schweizerische mineralogische und petrographische Mitteilungen) is also a chapter of Pawlig's PhD thesis from 2001; here is the link to the thesis:

<https://publications.ub.uni-mainz.de/theses/volltexte/2001/234/pdf/234.pdf>

In Pawlig and Baumgartner 2001 (their table 1 and fig. 5), the average silica content of the metagranite (70.24 wt%) and whiteschist (69.88 wt%) are the same within uncertainty ($\sigma_1 = 0.41$ for metagranite and 2.66 for whiteschist). Mass balance calculations show that there was no removal or addition of silica (as well as most other elements) in the metagranite (during metasomatism or later during alpine orogeny). The isocon data also confirm that the metagranite is the protolith of the whiteschist. This has been reconfirmed by data from the first author's PhD thesis. We do not discuss in detail in the text the granitic protolith and the nature of the metasomatic fluids because this issue is discussed in detail in the article of Pawlig and Baumgartner 2001 and their results have been confirmed by the thesis of Luisier. However, we added more information in the introduction of our manuscript in lines 50-52.

Deformation of whiteschist/metagranite and numerical model

The author's claim that the whiteschists are undeformed (l.46/47)

We disagree. We mentioned in line 47 (manuscript before revision) that the whiteschist is locally deformed: " local deformation in the whiteschist pods post-dates Alpine peak pressure.". We rephrased that sentence in order to clarify the timing of the deformation observed in lines 61-63. We mention in several places of our manuscript that essentially all deformation is retrograde, that is after the peak pressure conditions.

(which makes the use of the term "schist" quite awkward from a Structural Geology point of view),

The term whiteschist was defined by Schreyer back in 1974. The term refers to a specific mineral assemblage and not to the fabric of the rock. We added an explanation and reference to Schreyer's paper in line 38-40.

and that the "Metagranites closer than 80 meters to the whiteschist have not been analysed, since their high-pressure assemblage was strongly altered during a late greenschist-facies retrogression associated with deformation", meaning that the metagranite, at least close to the undeformed whiteschist is (strongly) deformed. How based on that can one assume "a simple two-dimensional analytical model in which the whiteschist is mimicked by an ellipse surrounded by a mechanically more competent material representing the metagranite"? To me it sounds as if the whiteschist is mechanically more competent and thus undeformed. Therefore saying in l.199-201: "Significant mechanical strength of the metagranite can be justified because field observations show that large volumes of the metagranite have remained essentially undeformed", and in l.210/211: "Therefore, the pressure gradient determined in the host metagranite could well reflect heterogeneous pressure distribution in the strong host" does not hold true, at least not close to the studied undeformed whiteschist on which the entire story hinges. Since the metagranite was infiltrated by metamorphic fluids and affected by associated deformation during greenschist-facies metamorphism, one cannot assume that the granite was strong, impermeable and undeformed during high-pressure metamorphism (l. 223/224). One simply does not know, at least not for the studied example.

Most of the metagranite is essentially undeformed. If there is deformation, it is greenschist facies deformation, that is deformation that occurred after the peak pressure. We clarified this point in lines 61-63.

There is simply no indication for a fluid event during the peak pressure stage. The major fluid event happened before the Alpine orogeny, during the hydrothermal alteration of the granite (whole rock and isotopes data presented in Pawlig and Baumgartner 2001, and the PhD thesis of Pawlig confirms this).

To me it appears very likely that the metagranite close to the whiteschist suffered deformation during the high-pressure stage, therefore was anisotropic allowing later fluid infiltration and deformation under greenschist-facies deformation, which then erased the earlier high-pressure tectonometamorphic features.

This scenario is a pure speculation of the reviewer and indeed a common interpretation which aims to be compatible with the lithostatic pressure paradigm. For the authors of this article, most of them having analyzed the mentioned rocks in the field, this scenario is not at all likely. There is no evidence for high, or peak, pressure deformation in the field and also not in the geochemical data. There is also no evidence that all peak-pressure features were erased. The transition zone between whiteschist and metagranite has a variable degree of deformation and in some places, there is essentially no deformation. There is no evidence, neither in the metagranite nor in whiteschist of a deformation during the high pressure. Our inclusion model applies to the peak-pressure conditions when the metagranite was essentially undeformed before the local greenschist facies deformation. We added text clarifying this on lines 258-262.

Whiteschist sheared in or not?

I realise that the author's make the point in l.35-36 that: "The whiteschists are not

xenoliths or pieces of the subduction channel that could have been incorporated later by tectonic mixing (mélange) into the granite during the Alpine orogeny" and have not been tectonically incorporated into the granite. They argue that this is because: "The absence of shear zones and faults related to the whiteschist outcrop imply tube-like chemical alteration structures (Fig. 1B) and the lack of any high-pressure foliation associated in the whiteschist" (l. 45-47). However, if the whiteschist would have been sheared into the surrounding metagranite then one would start looking for shear zones in the metagranite, not in the whiteschist. Judging from Fig. 1, the metagranite appears definitely deformed.

The outcrop conditions in the region allow for an excellent observation of the transition zone between the metagranite and whiteschist. The transition is gradual and marked by the progressive disappearance of igneous phases towards the whiteschist. The transition zone is discordant and lobate relatively to the (late alpine greenschist-facies) foliation. Numerous Permian-age (pre-Alpine) dykes crosscut the metagranite and whiteschist with no significant deformation and no direction change around the transition (indicated in our Fig. 1C). Hence, the whiteschists are in situ and not xenoliths or sheared-in rock units (see also discussion below).

Details were added in the introduction to clarify the in-situ character of the whiteschist in lines 47-50 and 56-63 and a field picture showing the metagranite-whiteschist transition was added in the appendix (Appendix 1).

Saying that: "...were generated from metasomatic alteration of the granite by late magmatic hydrothermal fluids related to the cooling of the granite intrusion long before the onset of Alpine orogenesis" is fine but does not exclude that they were sheared into the metagranite in the Alpine subduction zone.

While, in a general case, this might be indeed not possible to distinguish these scenarios, it is here in the case of the field area described in this study. Outcrop in the area is excellent. Many undeformed contacts are visible on outcrop scale. Indeed, this is demonstrated by the filed photo added to the appendix 1 (Figure A1). In addition, shearing on the granite-whiteschist boundary would disrupt the continuity of the dyke shown in Fig. 1C

The author's might want to read back to Gerya et al. TECTONICS, VOL. 21, NO. 6, 1056, doi:10.1029/2002TC001406, 2002, who discuss that:

"It may be suspected that the internal structure of orogenic belts with exposed high-pressure and ultrahigh-pressure metamorphic units, commonly referred to as nappes, with a thickness on the order of 10² to 10³ m and a horizontal width of exposure on the order of some 10³ to 10⁴ m, which feature different maximum P and T conditions, and contrasting P -T histories, may in fact partly represent material that has been extruded from a widening subduction channel to form an orogenic root beneath the forearc during ongoing subduction."

And also: "An array of diverse P-T paths rather than a single P-T trajectory is expected to be characteristic for subduction related metamorphic complexes. In fact, this appears to be what is commonly observed in orogenic belts. The characteristic size and shape of the units with an individual history depend on the rheology of the material in the subduction channel. Our simulations indicate that lower viscosities result in enhanced disintegration and mingling, and hence smaller characteristic length scales for coherent units and a marked contrasts between adjacent slices, a structure commonly termed melange, while higher viscosities favor the formation of extensive coherent slices,

possibly representing the type of structures commonly referred to as nappes in high-pressure metamorphic terranes.”

We know the articles of Gerya and co-authors, of course, very well. These early numerical models of Gerya et al. have been also nicknamed "washing machine" models because they predict a rotation, overturning and mixing of rock units within an orogeny. We do not, in a general case, deny the importance of this paper. Unfortunately, or rather fortunately for us, such behavior is totally inconsistent with the geological field evidence of the outcrops discussed here from the Monte Rosa nappe (see discussion concerning “whiteschist xenolith”) and the Alps in general, which are an ordered and imbricate nappe stack (as mentioned now in our revised introduction). We have modified the text to make this clear in line 30-32. We also would like to inform the reviewer that Gerya (Gerya, *Journal of Metamorphic Geology*, 2015) has published more recently a review article on tectonic pressure and deviations from lithostatic pressure in which he clearly supports the potential existence of significant deviations from lithostatic pressure.

It all hinges on whether the whiteschist they studied is in-situ or not. Because this is so critical for the argument in this manuscript I would think it may be a good idea to prove that more rigorously.

As outlined above, we have added additional information in line 47-61, including an image to proof our point in the appendix.

If the whiteschist would be a xenolith, formed at ca 80 km depth corresponding to 2.2 GPa lithostatic pressure, which was then mixed into the granite, formed at ca 50 km depth (1.4 GPa pressure), there would be ca 30 km difference in burial depth. There should be clear deformation features which would have accommodated the 30 km displacement to mix the whiteschist into the metagranite. The deformation intensity of the metagranite around the whiteschist is heterogeneous, however, observed shear zones are small, mostly retrograde and they accommodated a maximal displacement of a few tens of meters. Also, the crosscutting pre-Alpine dykes is a rigorous proof that the whiteschist is not a xenolith.

Minor comments

L.67-69: “Obviously, the lithostatic pressure assumption is crucial to all these arguments and understanding of metamorphic pressure variations recorded in the Monte Rosa nappe”. This is worded much too strongly as the higher-pressure units could have been sheared into the surrounding rocks in the subduction channel (see for instance Gerya et al., 2002, *Tectonics*, reference is given below). See above.

We do not agree, see response above.

L.135: “a plateau with decreasing Si-content away from the plateau” is pretty awkward wording.

We agree that "plateau" is not a common terminology. We reworded it as a bell shape profile lines 159-163 and in the figure caption.

L.54-56: the sentence “orogenic wedge model12... explains nappe formation and stacking by basal accretion and nappe exhumation by ongoing stacking, subsequent backfolding and erosion” does not appear plausible to me as in this scenario erosion is

assumed to be the sole exhumation mechanism. Dozens of tectonic studies have convincingly shown that this is not the case in this part of the Alps. Next sentence that buoyancy did not play a role. This might be true but there needs to be a reference!! Most people (not me!) worrying on exhumation of (U)HP rocks strongly believe that buoyancy is important.

In the original model of Platt, exhumation is a combination of local extension and erosion. Extension takes place in the upper part of the wedge. In the wedge model, the buoyancy does not play a major role. We added the reference to Platt (1986) line 72. However, buoyancy plays a major role in the subduction channel model (Butler et al. 2013, 2014).

L.63-64: "...subduction channel model¹³. This model explains nappes as buoyancy-driven plumes rising along a subduction channel from depths >80 km^{13,14}". This does unfortunately not make sense either as nappes are not plumes. This is absolutely wrong terminology.

We agree that nappes are not plumes and that the terminology is not appropriate. We just cite the terminology used in the interpretation of Butler et al. (2013, their figure 8b). They have interpreted the internal crystalline massif as high-pressure plumes. In fact, all subduction channel models imply that tectonic nappes are essential buoyancy driven plumes. We disagree with this interpretation and argue that tectonic nappes are thrust sheets formed within an orogenic wedge. We modified the text line 78 to make it clear.

L.75: Not sure whether "retrogression" is the proper term here as you apparently think the rocks were isothermally decompressed and did not cool during initial exhumation. Same for "retrograde" in l.105.

We think that retrogression is an appropriate term here because it refers to the retrograde path, i.e. the post-peak history.

L.99: Statements like "Interestingly, jadeite has never been reported in the literature..." need to be backed up by references (as you refer to "literature").

We added two references of studies done in the Monte Rosa metagranite who did not report jadeite, references are in line 115. None of the published studies focusing on the Monte Rosa nappe reported jadeite in the Monte Rosa nappe but we cannot mention all these studies here.

L.196/197: I think it would help a general audience if this was explained a bit more "mean stress of 1.4 GPa (centre of Mohr stress circle) and a differential stress of 1.4 GPa (diameter of Mohr stress circle)." The mean stress is $(\sigma_1 + \sigma_3)/2$ and the differential stress is simply $\sigma_1 - \sigma_3$. I did not do a full calculation but I think it means that a σ_1 of 2.1 GPa and a σ_3 of 0.7 GPa has been assumed. Please say how you arrived at those values. Why have they been chosen? What is the basis for these numbers?

The choice of the values of σ_1 and mean stress are: mean stress in the metagranite is 1.4 GPa, the differential stress is also 1.4 GPa so that the maximal principal stress, σ_1 , in the metagranite is 2.1 GPa in our model.

The model shows that the thermodynamic pressure in the weak whiteschist is close to σ_1 and the thermodynamic pressure in the metagranite is close to the mean stress. We also cite Jamtveit et al 2018, in which this model has been applied to shear zones in Western Norway and in this article, there is a sketch that illustrates the Mohr circle explanation.

The applied stress values are illustrative values which generate a pressure difference of 0.7 GPa (2.1 – 1.4 GPa). However, our estimated pressure difference has an error range from 0.5 to 1.1 GPa (0.8 plusminus 0.3 GPa) so that many other stress values are possible to generate such pressure difference. To clarify that section, we added details line 245-252.

L.197/198: "a compression direction of 75° with respect to the inclusion's long axis". This angle comes out of the blue. Why is such an angle chosen? This needs some explanation. (see my general comment above).

In fig. 6 we provide a systematic study showing that there is a range of orientations (70-90 degrees) and a range of aspect ratios (larger than 4) that result in a pressure of ca 2 GPa in the inclusion. The particular example employing 75° is to illustrate a representative example of the results of fig. 6. We rephrased line 242-244 to make it clear.

L.202/201: Saying that "field observations show that large volumes of the metagranite have remained essentially undeformed" does not help. Only important is the deformation of the metagranite at the contact with the studied whiteschist, and this metagranite appears deformed (Fig. 1), as you explicitly say in your text.

The metagranite is deformed locally and mostly retrograde, after peak pressure. The scarce deformation zones are distributed throughout the Monte Rosa granite. They are not concentrated around the whiteschist, nor are they spatially related. There is no indication at all of any deformation during the Alpine prograde or high pressure stages (see discussion above) and our model applies to the high pressure stage.

L.250/251: "Such burial depths agree with field-based structural restorations." Again one of those many statements that would need references. This is a recurring problem in the manuscript. There is simply incomplete documentation. I strongly assume that the restoration cannot estimate burial depths of nappes and that the latter have either been inferred or taken from field studies. If so, this is a circular argument.

We rephrased line 313.

L.255/256. Same here, "exhumation cycle of the Monte Rosa nappe and was supposedly short-lived ($\ll 1$ Ma)." This statement is absolutely not in line with what has been said in l.54-56. If one exhumes high-P rocks that extremely fast and is inferring erosion to be the main agent, then one needs erosion rates of $\gg 4$ cm/yr (current average erosion rates in the Alps are < 1 mm/yr).

The short-lived ($\ll 1$ Ma) refers to the short duration of the high differential stress episode. We rephrased the sentence to make it clear, line 321-323.

Stockholm 2 September 2018

Reviewer #2 (Remarks to the Author):

I am satisfied that the authors have adequately addressed the most of the reviewers' queries. A considerable amount of the reviewers' comments were misinterpretations and confusions due to the presentation of the previous version, and I believe the authors endeavored to convey their thoughts in the better-described Methods and more elaborate Discussion sections. Although the authors do not give quantitative answers to all the questions, these were indeed beyond the scope of this study and did not challenge the main conclusion of this study. I think the paper will be an excellent contribution to the metamorphic and Alpine communities, and the exciting story will stimulate further inquiries and studies. Nonetheless, I still have a few concerns that the authors might want to address before the manuscript is suitable for publication in a Nature journal.

I agree with the authors that the water activity in the metagranite is unlikely much smaller than unity, and the pressure estimate is not very sensitive to $a_{\text{H}_2\text{O}}$. However, the derivation, as the authors present it (Lines 316-353), creates confusion because we usually define the activity and μ_0 of the $\text{KAl}_3\text{Si}_3\text{O}_{10}(\text{OH})_2$ endmember instead of H_2O in muscovite. The physical meanings of " $\mu_0_{\text{H}_2\text{O}}^{\text{phen}}$ " and "activity of water in phengite" are vague. Also note that $X_{\text{H}_2\text{O}}^{\text{phen}}$ in the authors' formalism is not $\text{OH}/2$ (Line 283), but is $(\text{OH}/2)/(1 + \text{OH}/2 + F + \dots)$ in which "1" stands for the hypothetical $\text{KAl}_3\text{Si}_3\text{O}_{11}$ endmember. Even if we assume that Equation 3 is qualitatively correct, $a_{\text{H}_2\text{O}}^{\text{fl}}$ is not necessarily 1 when " $a_{\text{H}_2\text{O}}^{\text{phen}}$ " (which is approximated by $X_{\text{H}_2\text{O}}^{\text{phen}}$) is 1 (Line 342). In addition, their derivation also assumes fluid presence, which, the authors also agree, is possibly not the case (Line 153).

We have slightly changed the paragraph dealing with this in the "method" section" and moved it to the appendix (Appendix 2), hoping to clarify the approach chosen and why it was chosen. As we state, an actual quantitative calculation of water activity based on hydroxy and oxy-phengites would require experimental calibration. We have added some sentences in the appendix line 605-636 and the main paper section 103-106 and 196-197.

That said, I do not question the pressure estimate for metagranite. I believe Fig. 5 alone (without the grey highlighting) would suffice the purpose. The authors could remove the $a_{\text{H}_2\text{O}}$ derivations from the Method section, and replace $a_{\text{H}_2\text{O}}$ with X_{OH} in Figure 2. Too many assumptions made in the $a_{\text{H}_2\text{O}}$ estimate distract the readers and weaken the results.

As requested, we removed the grey box on figure 5. Indeed, we simplified the derivation, moved it in the appendix line 605-636 and use the water contents as "indicative", following the suggestion of the reviewer partially. See lines 609-610. In figure 2, we replaced " $a_{\text{H}_2\text{O}}$ " by " X_{OH} ".

Minor comments:

Lines 83-86. This sentence of analytical method should be removed from the Result section.

We removed that sentence as it is already mentioned in the method section.

Line 90. The results of metagranite could be presented in a new paragraph

We agree, we made a new paragraph.

Line 94. The "which ..." clause should be reworded, because "which" can also be interpreted as "the absence of jadeite".

We reworded to clarify the sentence. Line 110.

Line 114. larger -> higher?

Yes, we replaced larger by higher line 129.

Line 133. A few words seem missing: the proposition of WHAT?

There are words missing, it is the proposition of Evans and Patrick (1987). We added the missing words line 165.

Line 142. pressure increases ... -> the calculated pressure increases ...

OK

Line 151-152. "The activities are considerably lower than in the whiteschists." This sentence contradicts Line 157 "Water activities close to unity ..."

We rephrased to make it clear, line 194-195.

Line 156. Pressure gradient should be GPa/m, or the authors could replace "pressure gradient".

We rephrased line 201 and replaced "pressure gradient" by "pressure difference", line 209 and 265.

Line 168. "whiteschists" should be "metagranites"?

Here, "the calculated positions of reactions in whiteschists" refers to the thermodynamic calculations shown in fig. 2A, so it concerns the whiteschist, but we agree that this sentence needed to be clarified and to do so, we separated it into two sentences, presenting the uncertainty related to the metagranite and the uncertainty related to the whiteschist separately (line 211-212).

Line 178. "Ref 23" should be Ref 28?

yes, we corrected.

Line 249. What does "more moderate" mean? "More moderate" than what?

Yes, we agree that "more moderate" is subjective here and vague. We rephrased it lines 312-313.

Line 309. Please define all the variables. What do subscripts "1", "0" and "*" mean?

Definition was added line 381-382.

Reviewers' comments:

Reviewer #4 (Remarks to the Author):

I have been asked to give my opinion about the exchange between R#1 and the authors of the manuscript "Metamorphic pressure variation in a coherent Alpine nappe challenges lithostatic pressure paradigm". After carefully reading the manuscript and the exchange between R#1 and the authors I conclude that the authors have a point and the manuscript should be published. Let me elaborate my opinion in more detail. Identifying pressure variations in metamorphic rocks are at the fore-front of metamorphic and structural geology, although critically debated. The authors present a field example of a whiteschist rocks within a metagranite where geochemical analysis suggest pressure variations of 0.8 GPa. The main argument between R#1 and the authors is whether the whiteschist represents an in-situ structure or has been exhumed from greater depth as R#1 argues. Looking at Figure 1c and Appendix A1 I do not see any geologically justified reason why the whiteschist should be a xenolith within a subduction zone channel. Figure 1 c shows dykes cross-cutting the metagranite and the whiteschist and most of the contact between the two rock types is undeformed. Thus, I agree with the authors that these rocks represent an in-situ situation. Having a look at the mineral chemistry analysis and the thermodynamic calculations everything appears to be internally consistent. I would like to ask the author to, however, include error bars in Figure 4 B and C to indicate the robustness of the microprobe analysis. An additional point that was raised by R#1 is that the increase in Si content is due to a metasomatic effect. The authors disagree with this interpretation by citing Pawlig and Baumgartner 2001, referring to the isocon analysis in Fig 5. Having looked at this paper, that the authors attached to their review, I agree that there is no evidence for any metasomatism between the granite and the whiteschist, specifically not for Si. As this is a critical point for the manuscript and it appears that R#1 seemed to have trouble getting to the Pawlig and Baumgartner 2001 paper a general reader may also have trouble getting this paper. Why not use the data in paper and replot the isocon diagrams in the current manuscript's appendix? This way any misconceptions can be avoided. I leave this up to the authors to decide but I feel it would be the best way forward. Moreover, could the authors make a back of the envelope calculation how much Si would be needed to increase the Si content within the phengite and compare this to the isocon analysis? This would rule out any doubts. Saying this, looking at Fig 5 in Pawlig and Baumgartner I see that a best fit isocon was used. How would this change assuming constant volume? Would there be any reason to assume mass balance analysis based on constant volume is justified? The replacement of plagioclase by albite appears to be pseudomorphic so no massive solid volume change in the granite. If justified would it change the mobility of Si? An additional minor comment, it appears that the absence of jadeite is important in the whole argumentation chain. The authors nicely show that the albite is a replacement product of the plagioclase which is well demonstrated in Fig 3 A and B. To make the case stronger that albite cannot be a reaction after jadeite but a replacement after plagioclase the authors could cite Plümer et al. 2017 Nature Geoscience, who show identical microstructures in Fig 2, in for example line 123 of the manuscript. In conclusion, given the provocative nature of the manuscript, the compelling evidence that the whiteschist and the meta-granite represent an in-situ situation and the robust geochemical

analysis (if the isocon diagram may well be added to the appendix) I would like to see this paper published in Nature Communications. It will greatly add to discussions about pressure variations that are at the fore-front of current research in solid Earth sciences.

Utrecht, 20.05.2019

Oliver Plümper

Reviewer #4 (Remarks to the Author):

I have been asked to give my opinion about the exchange between R#1 and the authors of the manuscript "Metamorphic pressure variation in a coherent Alpine nappe challenges lithostatic pressure paradigm". After carefully reading the manuscript and the exchange between R#1 and the authors I conclude that the authors have a point and the manuscript should be published. Let me elaborate my opinion in more detail. Identifying pressure variations in metamorphic rocks are at the fore-front of metamorphic and structural geology, although critically debated. The authors present a field example of a whiteschist rocks within a metagranite where geochemical analysis suggest pressure variations of 0.8 GPa. The main argument between R#1 and the authors is whether the whiteschist represents an in-situ structure or has been exhumed from greater depth as R#1 argues. Looking at Figure 1c and Appendix A1 I do not see any geologically justified reason why the whiteschist should be a xenolith within a subduction zone channel. Figure 1 c shows dykes cross-cutting the metagranite and the whiteschist and most of the contact between the two rock types is undeformed. Thus, I agree with the authors that these rocks represent an in-situ situation. Having a look at the mineral chemistry analysis and the thermodynamic calculations everything appears to be internally consistent. I would like to ask the author to, however, include error bars in Figure 4 B and C to indicate the robustness of the microprobe analysis.

The uncertainty on the microprobe analyses of silica is between 0.1 - 0.3 % (relative). We have put error bars corresponding to the maximum error of 0.3% (relative) of the silica composition and this resulted in error bars that are smaller than the symbol size on the figure. Hence, they are not visible. Therefore, the tendency displayed by the profiles in Fig. 4b and 4c is robust. We added a sentence in the figure caption in lines 580-581.

An additional point that was raised by R#1 is that the increase in Si content is due to a metasomatic effect. The authors disagree with this interpretation by citing Pawlig and Baumgartner 2001, referring to the isocon analysis in Fig 5. Having looked at this paper, that the authors attached to their review, I agree that there is no evidence for any metasomatism between the granite and the whiteschist, specifically not for Si. As this is a critical point for the manuscript and it appears that R#1 seemed to have trouble getting to the Pawlig and Baumgartner 2001 paper a general reader my also have trouble getting this paper. Why not use the data in paper and replot the isocon diagrams in the current manuscript's appendix? This way any misconceptions can be avoided. I leave this up to the authors to decide but I feel it would be the best way forward.

We agree with the reviewer and added the Isocon calculated with the data from Pawlig and Baumgartner in the appendix (Appendix 3).

Moreover, could the authors make a back of the envelope calculation how much Si would be needed to increase the Si content within the phengite and compare this to the isocon analysis? This would rule out any doubts.

We understood from the comments of reviewer #1 that the metasomatism could be misleadingly associated to the spatial variation in silica in the metagranite samples. Therefore, the Isocon in appendix 3, as well as the related explanation in the main text in lines 179 to 182 present data showing that there is no silica enrichment from the metasomatic fluid. However, thermodynamically, the silica content in phengite is not dependent on the silica concentration of the rock as long as quartz is present in the mineral assemblage. The presence of quartz fixes the chemical potential of SiO_2 to 1 and defines the silica content in phengite.

Saying this, looking at Fig 5 in Pawlig and Baumgartner I see that a best fit isocon was used. How would this change assuming constant volume? Would there be any reason to assume mass balance analysis based on constant volume is justified? The replacement of plagioclase by albite appears to be pseudomorphic so no massive solid volume change in the granite. If justified would it change the mobility of Si?

Pawlig and Baumgartner used the least square approach of Baumgartner and Olsen (1995). Actually, the reason to use an Isocon Diagram is exactly that one does not need to assume constant volume, nor a priori immobile elements. The mass transport solution is found by finding the solution which leaves the majority of elements immobile. This is a reasonable assumption for rocks undergoing only minor alteration. It can be further elaborated by using geochemical reasoning about the immobility of high field strength elements. See Baumgartner and Olsen, 1995 for further discussions.

An additional minor comment, it appears that the absence of jadeite is important in the whole argumentation chain. The authors nicely show that the albite is a replacement product of the plagioclase which is well demonstrated in Fig 3 A and B. To make the case stronger that albite cannot be a reaction after jadeite but a replacement after plagioclase the authors could cite Plümper et al. 2017 Nature Geoscience, who show identical microstructures in Fig 2, in for example line 123 of the manuscript.

We added the reference to Plümper 2017 in line 124, as suggested by the reviewer.

In conclusion, given the provocative nature of the manuscript, the compelling evidence that the whiteschist and the meta-granite represent an in-situ situation and the robust geochemical analysis (if the isocon diagram may well be added to the appendix) I would like to see this paper published in Nature Communications. It will greatly add to discussions about pressure variations that are at the fore-front of current research in solid Earth sciences.

Utrecht, 20.05.2019

Oliver Plümper

REVIEWERS' COMMENTS:

Reviewer #4 (Remarks to the Author):

I have now received the answers to my review and I am very happy with the work the authors have done to answer my queries. I do not have any further comments and would like to suggest that the paper is accepted. Congratulations on a provocative and innovative publication that presents a valuable discussion of non-lithostatic pressure variations.